# Efficient industrial-current-density acetylene to polymer-grade ethylene via hydrogen-localization transfer over fluorine-modified copper

Lei Bai[1], Yi Wang [1], Zheng Han[1], Jinbo Bai[2], Kunyue Leng [1] ✉, Lirong Zheng[3] ✉, Yunteng Qu [1] ✉ & Yuen Wu [4]

Electrocatalytic acetylene semi-hydrogenation to ethylene powered by renewable electricity represents a sustainable pathway, but the inadequate current density and single-pass yield greatly impedes the production efficiency and industrial application. Herein, we develop a F-modified Cu catalyst that shows an industrial partial current density up to $0.76 \, A \, cm^{-2}$ with an ethylene Faradic efficiency surpass 90%, and the maximum single-pass yield reaches a notable 78.5%. Furthermore, the Cu-F showcase the capability to directly convert acetylene into polymer-grade ethylene in a tandem flow cell, almost no acetylene residual in the production. Combined characterizations and calculations reveal that the $Cu^{\delta+}$ (near fluorine) enhances the water dissociation, and the generated active hydrogen are immediately transferred to $Cu^0$ (away from fluorine) and react with the locally adsorbed acetylene. Therefore, the hydrogen evolution reaction is surpassed and the overall acetylene semi-hydrogenation performance is boosted. Our findings provide new opportunity towards rational design of catalysts for large-scale electrosynthesis of ethylene and other important industrial raw.

Ethylene ($C_2H_4$), as a primary building block for polyethylene, meets a vast annual consumption[1]. At present, industrial-scale production of $C_2H_4$ is still dominated by the petroleum cracking, which needs consume the finite petroleum reserves accompanying with high energy consumption[2,3]. Recently, coal-derived acetylene followed by thermal semihydrogenation (SAE) is recognized as a promising route to $C_2H_4$ production[4–6]. However, the SAE process suffers from excess hydrogen consumption, high reaction temperature, and pressure[7–9]. Furthermore, an additional acetylene removal step may need to satisfy the stern requirement of acetylene content in polymer-grade ethylene (not exceeding 5 ppm). Electrocatalytic SAE (ESAE) powered by renewable

electricity provides a sustainable alternative route to directly transform acetylene to polymer-grade ethylene[10–13]. Moreover, ESAE process adopts water ($H_2O$) as the hydrogen source, avoiding the consumption of $H_2$. Nevertheless, the severe hydrogen evolution reaction (HER) at industrial-level current density greatly restricts the Faraday efficiency (FE) and the one-pass yield of $C_2H_4$. In this case, the large recycle ratio is required to completely convert $C_2H_2$ to $C_2H_4$, thus greatly increase the production cost and compress the profit margin of ESAE. Actually, a fundamental demand in becoming ESAE profitable is a 85% or more FE of $C_2H_4$ at a current density of $0.2 \, A \, cm^{-2}$ or more[14]. Therefore, the rational design and fabrication of catalysts that can

[1]International Collaborative Center on Photoelectric Technology and Nano Functional Materials, Institute of Photonics and Photon-Technology, Northwest University, 710069 Xi'an, Shaanxi, China. [2]Université Paris-Saclay, CentraleSupélec, ENS Paris-Saclay, CNRS, LMPS-Laboratoire de Mécanique Paris-Saclay, 8-10 rue Joliot-Curie, Gif-sur-Yvette 91190, France. [3]Institute of High Energy Physics, 100039 Beijing, China. [4]School of Chemistry and Materials Science, University of Science and Technology of China, 230026 Hefei, China. ✉e-mail: lengky@nwu.edu.cn; zhenglr@ihep.ac.cn; yuntengqu@nwu.edu.cn

suppress HER and efficiently produce ethylene at industrial-level current densities is urgently desired.

The research on ESAE can be traced back to the 1970s[15], which was limited by poor solubility of acetylene and insufficiently effective catalysts. Very recently, Cu-based catalysts have been demonstrated to be prospective candidates for realizing the selective electrocatalytic semihydrogenation of acetylene[16–18]. Moreover, the issue of solubility has been addressed by utilizing gas diffusion layer (GDL) electrodes, which can improve mass transfer by creating a gas–liquid–solid interface[19–21]. However, the competing HER still greatly reduce the FE of $C_2H_4$ once the current density exceed $0.1\,A\,cm^{-2}$. In this regard, nanoscale copper catalyst (5 nm) with unsaturated Cu sites were developed and performed strong suppressive HER (< 4%) during ESAE at industrial-level current density ($0.2–0.5\,A\,cm^{-2}$)[22]. Despite significant progress has been achieved, the precise control of the size and crystal structure of nanoscale Cu catalysts may increase potential materials cost during industrial application of ESAE. Inspired by catalyst designed for $CO_2$ electroreduction reaction, modulating the local electronic structure of copper by non-metal modifying can efficiently boost reaction kinetics[23–25]. Furthermore, it has been demonstrated the introduction of non-metal, such as fluorine, boron, into Cu achieved robust suppressive HER performance at an industrial-level current density (surpass $1\,A\,cm^{-2}$)[26–29]. Therefore, reasonably constructing non-metal modifying Cu catalysts onto a GDL-coated carbon paper electrode are supposed to realize the industrial application of ESAE but remains challenging.

Herein, we develop halogen-doped Cu catalysts through an in situ electrical reduction method for achieving efficient ESAE process. Detailed physical characterizations of the catalysts reveal that halogen atoms are adsorbed onto the Cu (111) plane through halogen-Cu bonding. Among these catalysts, the F-doped catalyst (Cu–F) demonstrate superior performance in ESAE when working in the flow cell. The current density reaches a remarkable $1\,A\,cm^{-2}$ at $-1.5\,V$ (vs. RHE), and the FE of ethylene surpass 90% at a broad potential range from $-0.5$ to $-1.3\,V$ (vs. RHE). Notably, Cu–F showcase the capability to directly convert acetylene (70 mol% in Ar) into polymer-grade ethylene in a tandem flow cell device. The experimental verification combined with the DFT calculations reveals the boosting of water dissociation on the Cu sites near the F atom, resulting the generation of active hydrogen, which are transferred to the Cu far from F and react with the adsorbed acetylene. Thus, the HER is suppressed and the overall ESAE is promoted over Cu–F.

## Results

### Preparation and characterization of Cu–F catalyst

In this study, the electrocatalyst is synthesized by in situ electrical reduction using halogen-containing copper as a precursor (Fig. 1a). Typically, the preparation of the Cu–F involves treating $Cu(NO_3)_2$ and $NH_4HF_2$ in DMF via a solvent-thermal process, resulting in a green powdery precursor of Cu(OH)F (Supplementary Figs. S1–S3). Then the Cu(OH)F is coated onto a GDL and reduced at a potential of $-1.6\,V$ (vs. Ag/AgCl). The resultant Cu–F catalysts are separated from the GDL for further characterization. SEM and TEM images of Cu–F reveal the aggregation of irregular nanoparticles with a size ranging from 45 to 90 nm, which exhibit distinct crystalline characteristics (Fig. 1b and Supplementary Fig. S4). The EDS element mapping confirms the uniform distribution of F on the Cu particles (Fig. 1c), with the F content of 15.6 atomic% (Supplementary Table S1). Aberration-corrected HAADF-STEM is used to identify the surface structure of Cu–F. Figure 1d clearly illustrates the presence of Cu atomic alloy in Cu–F, with a lattice spacing measured at 0.208 nm corresponding to the Cu (111) plane. This observation is further confirmed by characteristic diffraction peaks of metallic Cu in Fig. 1e. Moreover, no obvious lattice distortion caused by heteroatomic

insertion is observed in HAADF-STEM image, combined with the F $1s$ XPS spectra (Supplementary Fig. S5), indicating the adsorption of F onto the surface crystalline Cu.

The linear sweep voltammetry (LSV) curve of Cu(OH)F reduction shows a peak at $-0.35\,V$ vs. RHE (Supplementary Fig. S6), indicating the occurrence of the electroreduction on $Cu^{2+}$ during the Cu–F generation. The Cu $2p$ XPS spectra (Supplementary Fig. S7) further confirm that Cu–F exhibits a binding energy for Cu $2p_{3/2}$ at 932.7 eV, slightly higher than that reported 932.4 eV for $Cu^0$[30]. In addition, the Cu LMM spectra (Supplementary Fig. S8) indicate the coexistence of $Cu^0$ and $Cu^{\delta+}$ in the Cu–F catalyst. X-ray absorption spectroscopy (XAS) was used to further investigate the oxidation state and coordination of Cu in Cu–F. The XANES spectra at the Cu $K$-edge indicate that the oxidation state of Cu in Cu–F is between 0 and +1 (Fig. 1f). The linear simulation of edge energy and oxidation state provides insight into the average oxidation state of Cu in Cu–F, which was approximately +0.2 (Supplementary Fig. S9). This slight positive oxidation state could be attributed to the presence of $Cu^{\delta+}$ induced by the interaction of Cu and F. The combination of Cu and F is further explored by Cu $K$-edge FT EXAFS in Fig. 1g, which showed a dominant peak at -2.19 Å corresponding to the Cu–Cu coordination. In addition, a peak at -1.56 Å is observed over Cu–F, situates between the Cu–O coordination in CuO and the Cu–F coordination in $CuF_2$. This observation in agreement with the wavelet-transform images (Fig. 1h–j and Supplementary Fig. S10), indicating the partial coordination of Cu with F in the Cu–F catalyst. Furthermore, the pure crystalline Cu (Cu NP) and Cl, Br, I-doped Cu catalysts (Cu-Cl, Cu-Br, and Cu-I) are also prepared and investigated in this work (Fig. 1e and Supplementary Figs. S11–14).

### Electrocatalytic acetylene semihydrogenation

The ESAE evaluation is conducted by a flow cell device with an electrode area of $1\,cm^{-2}$ (Supplementary Fig. S15). Subsequently, 1 M KOH and $C_2H_2$/Ar gas (70 mol%) are pumped into the flow cell respectively at the flow rate of 1 and $30\,mL\,min^{-1}$. (The addition of argon to the $C_2H_2$ gas was intended to protect the chromatographic system.). To ensure consistency, all potentials used in the experiment were converted to the RHE reference without solution resistance compensation, except the long-term stability test of Cu–F.

As disclosed by the LSV curves (Fig. 2a), Cu-based catalysts show obviously higher current density when exposed to $C_2H_2$ than to Ar, confirming their intrinsic activity for ESAE. Additionally, the halogen doping noticeably enhances the ESAE performance of Cu, with the current density increasing along the electronegativity. As result, the Cu–F shows the best ESAE performance with the current density surpasses $1\,A\,cm^{-2}$ at $-1.5\,V$ (vs. RHE). The ECSA-normalized current densities follow the same trend with the apparent ESAE performance (Supplementary Figs. S16 and S17 and Supplementary Table S2), ruling out the effect of the morphology, and demonstrating the superior intrinsic activity of Cu–F. The single-pass $C_2H_2$ conversion over Cu–F reaches 30.3 %, which increases to 78.5% when reducing the flow rate to $6\,mL\,min^{-1}$, and the LSV curves show no obvious change with the flow rate (Supplementary Fig. S18). Ethylene ($C_2H_4$) is identified as the main production, accompanied by few $H_2$ and $C_4$, no liquid production is detected (Supplementary Figs. S19 and 20). The partial current density for $C_2H_4$ is indicated up to $794.8\,mA\,cm^{-2}$ at the potential over $-1.5\,V$ (vs. RHE, Fig. 2b). Moreover, the Cu–F shows a remarkable onset potential of $-0.121\,V$ (at $0.1\,mA\,cm^{-2}$) for $C_2H_4$ (Supplementary Fig. S21), $-0.607\,V$ greater than that for the competing HER, demonstrating no external $H_2$ is required for producing $C_2H_4$. The notable $C_2H_4$ selectivity of Cu–F in alkaline medium is revealed by the high Faraday efficiency (FE), which surpass 90% at a wide potential range from $-0.5$ to $-1.3\,V$ (vs. RHE, Fig. 2c and Supplementary Fig. S22), better than the Cu NP (-80%) and other halogen-doped Cu (Supplementary Figs. S23–26). In the same

potential range, the highest $H_2$ FE over Cu–F is just 4.1% (−1.3 V), but for Cu NP it reaches 28.9% (−1.0 V). The superior current density and $C_2H_4$ selectivity lead to a high $C_2H_4$ formation rate of 1 mol h$^{-1}$ cm$^{-2}$ at −1.0 V (vs. RHE, Fig. 2d), making it as one of the best catalysts for ESAE (Supplementary Table S3). It is worth noting that ESAE performance of Cu–F negligible decay in 43 h at a current density of 200 mA cm$^{-2}$ (Fig. 2e and Supplementary Fig. S27), and the used Cu–F maintain the original structure (Supplementary Fig. S28), demonstrating its robust long-term stability. It is inevitable that part of F will leach into the electrolyte during the in situ electrical generation of Cu–F from Cu(OH)F. Supplementary Fig. S29 shows the ESAE performance of Cu–F in a fresh electrolyte (F-free), the negligible changes in the activity exclude the effect of the solvated fluoride on the ESAE process.

## Mechanistic insight

The nature of the active sites in Cu–F during the ESAE process is investigated by in situ XAFS (Supplementary Fig. S30). The freshly generated Cu–F (5 min in Ar) shows an average oxidation state of Cu between 0 to +1 with Cu-Cu and F-Cu coordination, confirming the innate Cu$^+$ sites accompany Cu$^0$ sites. Moreover, after triggering the ESAE by switch the gas flow to $C_2H_2$ for 120 min, the average oxidation state and coordination environment of Cu show slight changes, suggesting the maintaining of the F–Cu interaction. Considering the much more positive onset potential of ESAE (−0.12 V, Supplementary Fig. S21) than the reduction potential of Cu$^{\delta+}$ (−0.23 V, Supplementary Fig. S6), $C_2H_2$ is much easier to be reduced than Cu$^{\delta+}$. Therefore, the F–Cu coordination is protected from being deeply reduced, which contributes to the superior ESAE performance of Cu–F.

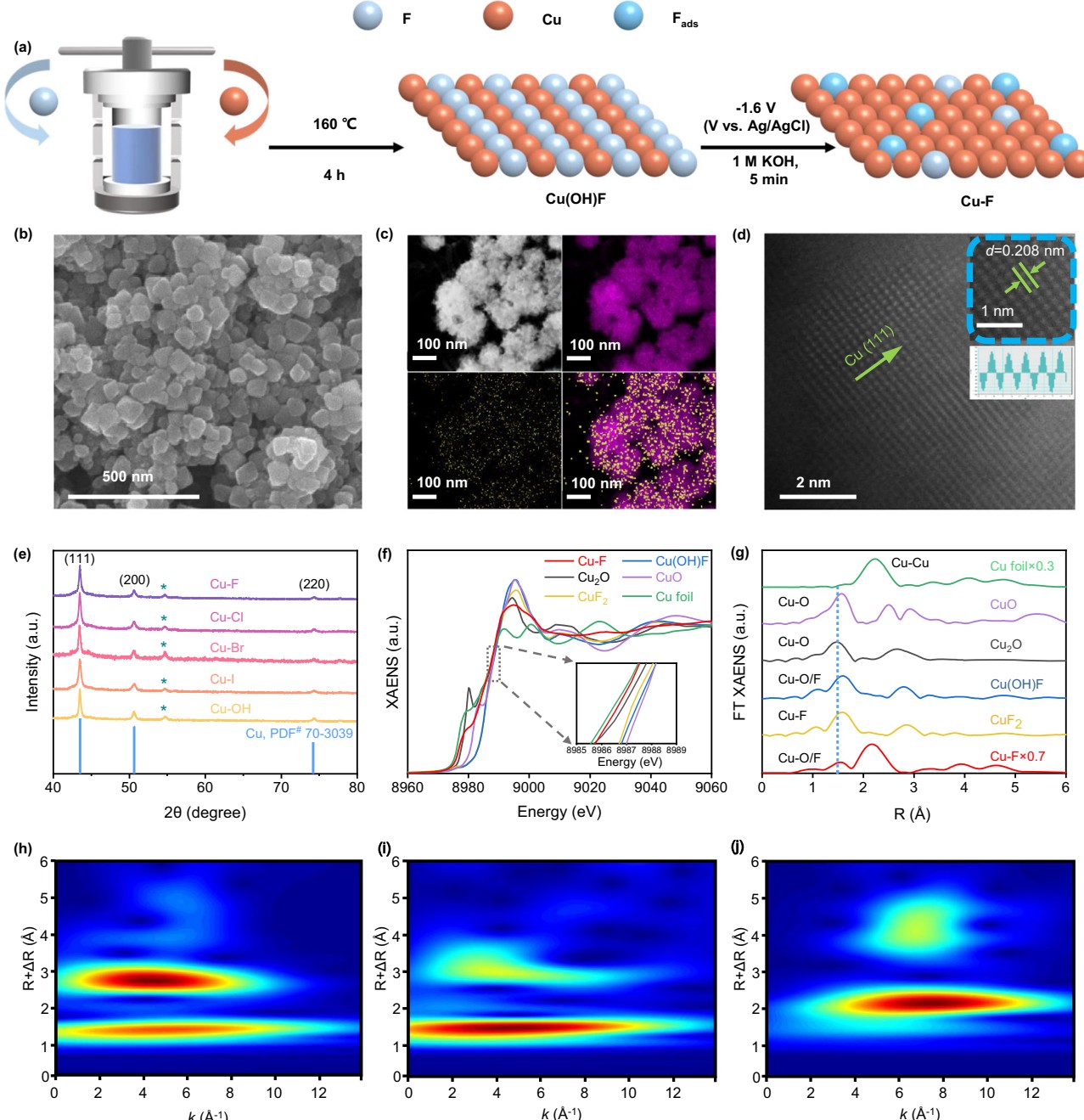

**Fig. 1 | Structure identification of Cu–F. a** Illustrated synthesis process. **b** SEM image. **c** EDS element mapping. **d** Aberration-corrected HAADF-STEM image. **e** XRD patterns. **f** XANES spectra with reference samples. **g** FT EXAFS spectra with reference samples. **h–j** Wavelet transformations of Cu$_2$O, CuF$_2$, and Cu–F.

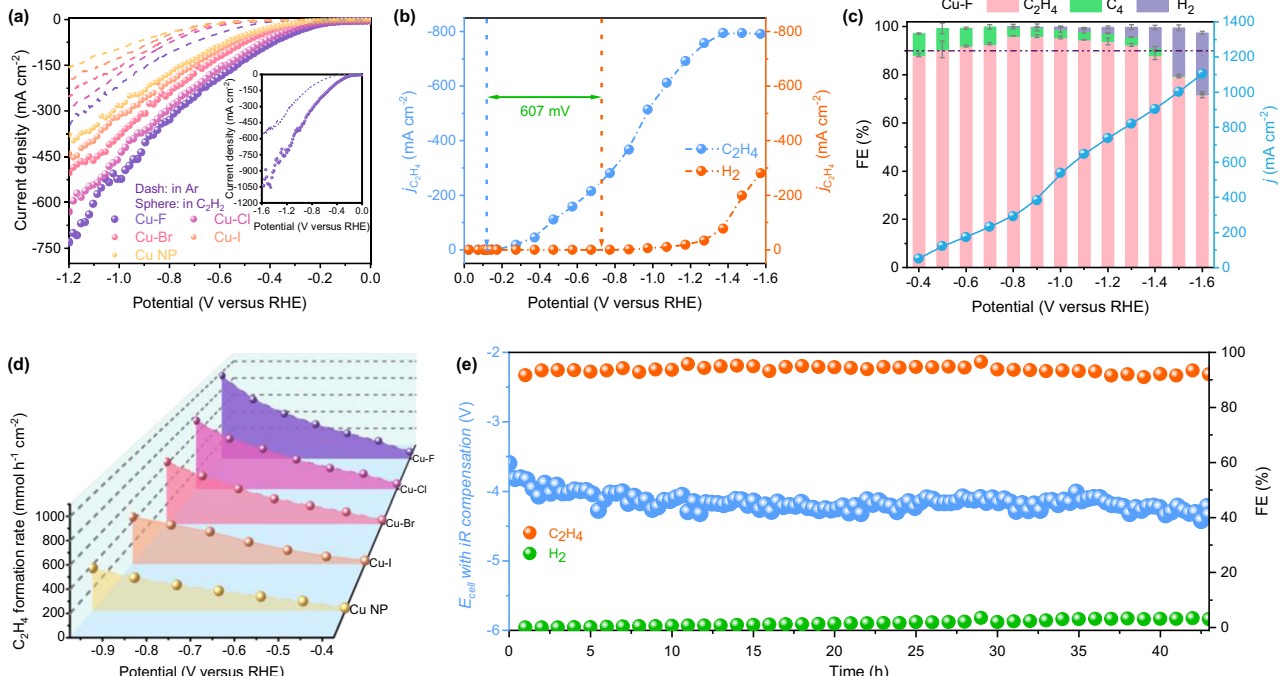

**Fig. 2 | Electrocatalytic performance of Cu–F in 1 cm² flow cell. a** LSV curves in Ar and 70% $C_2H_2$/Ar. **b** partial current density for $C_2H_4$ and $H_2$. **c** Faraday efficiency for $C_2H_4$ and $H_2$ at various potential and the corresponding current density, the maximum measurement error is ±5%. **d** $C_2H_4$ formation rate. **e** Long-term stability at a constant current density of 200 mA cm⁻². All tests are measured using a three-electrode flow cell (1 cm²) in 1 M KOH at room temperature under 70 mol% $C_2H_2$/Ar flow (30 mL min⁻¹). The results are presented without iR compensation except (**e**). For (**e**), the $R = 9.4\ \Omega$.

To elucidate the mechanism underlying the superior performance of Cu–F, ESAE pathway is first studied using potential-dependent operando Raman spectra (Supplementary Fig. S31). As shown in Fig. 3a, the two peaks at 1127 and 1513 cm⁻¹ can be assigned to the C-C and C≡C vibrations of polyacetylene[22]. The signal belonging to the C≡C vibration is detected at ~1700 cm⁻¹ at an open circuit potential (OCP), indicating the adsorption of acetylene on Cu–F[18]. This signal disappears when the potential increase to −0.4 V (vs. RHE), and new signals belonging to bonded ethylene emerge at ~1343 and ~1554 cm⁻¹ (see refs. 14,18), which confirm the occurrence of ESAE process on Cu–F. Furthermore, the signals of ethylene on Cu–F can be observed at wide potential range from −0.4 to −1.0 V (vs. RHE). In contrast, the Raman signal for ethylene over Cu NP becomes indistinguishable at a potential larger than −0.8 V (vs. RHE, Fig. 3b), which should be attributed to the intense hydrogen generation at high potential. Interestingly, the LSV curves in pure Ar show that Cu–F exhibits better HER activity compared to Cu NP (Fig. 2a). However, the FE of $H_2$ over Cu–F is much lower than that over Cu NP, suggesting the suppression of HER over Cu–F in ESAE process due to rapid *H consumption.

The pH of the electrolyte is investigated to reveal the hydrogen source in the ESAE. As reported by Deng et al., a typical electron-coupled proton transfer process (*$C_xH_y$ + $H_2O$ + e⁻ → *$C_xH_{y+1}$ + OH⁻) is more favorable under relative low pH[20]. However, the pH shows insignificant effect on the overall FE of Cu–F (Supplementary Figs. S32–S34), which excludes the electron-coupled proton transfer process. The hydrogenation of $C_2H_2$ is preferentially facilitated by the surface absorbed *H, which stem from the water dissociation. The surface *H is further judged by adding of *tert*-Butanol (*t*-BuOH) into the electrolyte. The *t*-BuOH can capture *H to form inert 2-methyl-2-propanol radicals, leading to a suppressed hydrogenation activity[31,32]. As shown in Supplementary Fig. S35, the Cu–F displays inferior performance after adding *t*-BuOH, confirming the participation of surface *H in the hydrogenation of $C_2H_2$. To gain insights into the role of water dissociation in *H generation, the kinetic isotope effect (KIE) of H/D in

studied during ESAE. The Cu NP shows a KIE of 2.01 (Fig. 3c and Supplementary Fig. S36), which is characteristic of primary KIE, indicating the involvement of water dissociation in the rate-determining step (RDS)[29]. For Cu–F, the KIE decrease to 1.26, water dissociation is no longer involved in the RDS, demonstrating that Cu–F accelerates the hydrogen transfer process via promoting water dissociation[33].

The effect of hydrated cation M⁺$(H_2O)_n$ (n represent the number of hydrations) in electrolyte is investigated to deeply insight the active sites for water dissociation. The hydrated cation would interact with the surface F in the Helmholtz layer, contributing to the dissociation of $H_2O$ to form active hydrogen species, and this interaction is determined by the n and radius of the hydrated cation[34]. As shown in Fig. 3d and Supplementary Fig. S37, when replacing 1 M KOH by 1 M tetramethylammonium hydroxide (TMAH) or NaOH, both the $C_2H_2$ conversion and HER over Cu–F exhibit significant decay, which due to the weaker interaction of F with TMAH and Na⁺$(H_2O)_{13}$ than with K⁺$(H_2O)_7$, caused by their larger radius and n[29,34]. In contrast, no obvious change on $C_2H_2$ conversion and HER over Cu NP are observed, although cations with different nature (such as different diffusion coefficient) are used (Supplementary Fig. S38). The investigation of hydrated cation reveals the key role of F doping for regulating water dissociation.

The DFT calculation is employed to further understand the ESAE process on Cu–F. In order to clarify the role of F atom, three different active sites are simulated, Cu^δ⁺ site near the F atom in Cu–F (Cu(111)-F-near), Cu⁰ site far from the F atom in Cu–F (Cu(111)-F-far) and Cu⁰ sites in pure crystalline copper (Cu (111)) (Supplementary Figs. S39–S40). The adsorption energy ($E_{ad}$) of $C_2H_2$ and $H_2O$ are first studied. As shown in Fig. 3e, the $E_{ad}$ of $C_2H_2$ on Cu (111)-F-near is much higher than that on Cu (111)-F-far, indicating the $C_2H_2$ is more favorable to be adsorbed on the Cu sties far form F atom than the Cu nearby. On the other hand, Cu (111)-F-near shows priority adsorption of $H_2O$ versus $C_2H_2$ confirmed by its lower $E_{ad}$ for $H_2O$ than $C_2H_2$. As results, the strong interaction between *$H_2O$ and Cu (111)-F-near significantly

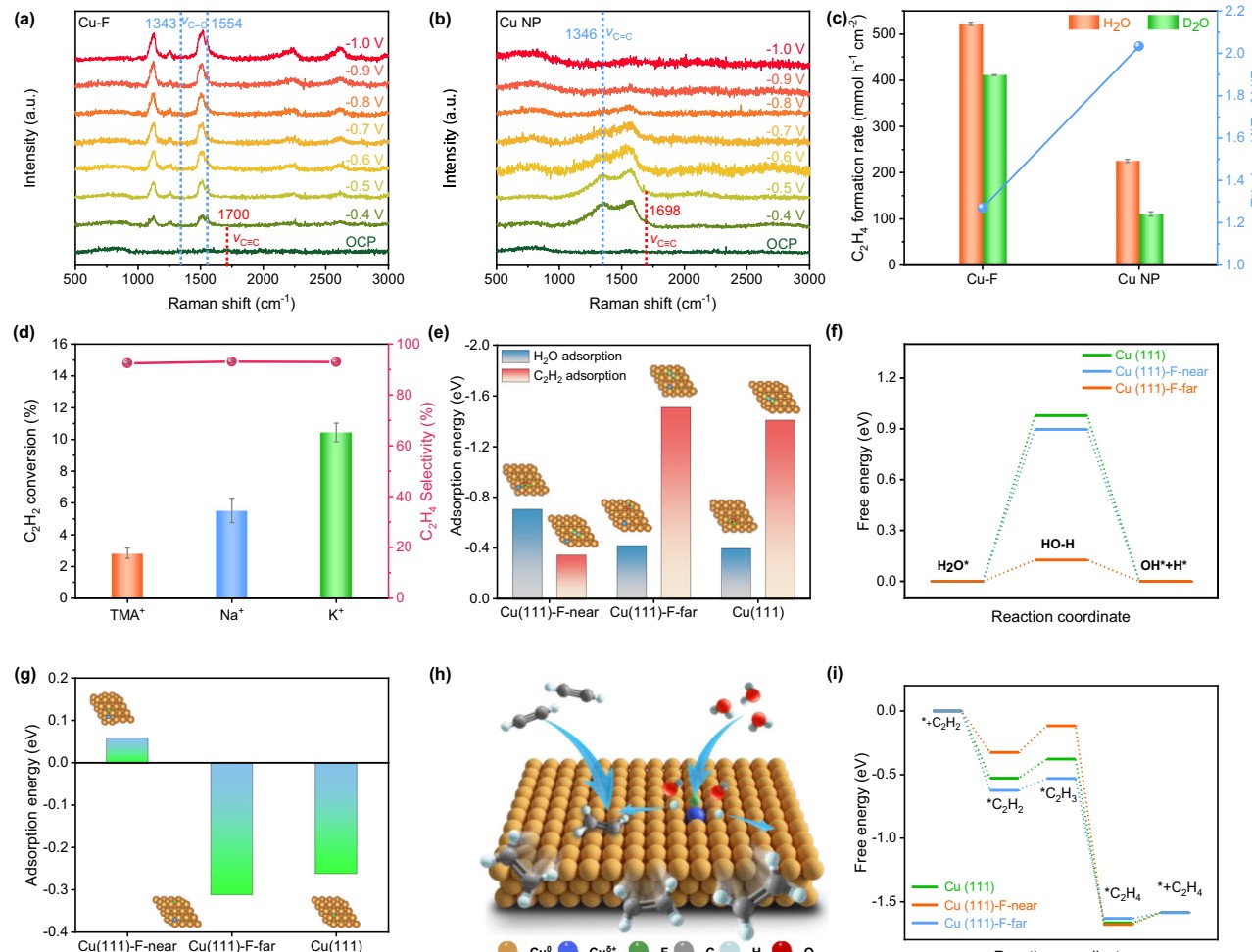

**Fig. 3 | Mechanistic insight. a, b** Potential-dependent operando Raman spectra, measured using a three-electrode observable cell in 1 M KOH at room temperature under 70 mol% $C_2H_2$/Ar flow (30 mL min⁻¹). **c** Kinetic isotope effect, KIE, the maximum measurement error is ±4.8%. **d** $C_2H_2$ conversion under 70 mol% $C_2H_2$/Ar flow (30 mL min⁻¹) in 1 M KOH, NaOH, and TMAH electrolyte. **e** Adsorption energy for $C_2H_2$ and $H_2O$, **f** free energy diagram for water dissociation and **g** adsorption energy for hydrogen, at 0 V vs. RHE. **h** Schematic illustration of the ESAE pathway over Cu–F. **i** Free energy diagram for the hydrogenation of $C_2H_2$ at 0 V vs. RHE.

promote the water dissociation (Fig. 3f), and boost the generation of active H species. Furthermore, the unfavorable ˙H adsorption on Cu (111)-F-near may enable the H transferring to Cu (111)-F-far (Fig. 3g), the low H transfer energy barrier of 0.105 eV further confirms the ease of transferring (Supplementary Fig. S41). Accordingly, an ESAE process includes hydrogen-localization transfer can be illustrated in Fig. 3h. Namely, $H_2O$ first absorb and dissociate on the Cu (111)-F-near, then the generated active H species are transferred to the Cu sites far from the F atom and react with the adsorbed $C_2H_2$, ultimately generate the $C_2H_4$ production. As shown in Fig. 3i, the rate-determining step for the as illustrated $C_2H_2$ hydrogenation process on Cu sites is identified as the hydronation of ˙$C_2H_2$ to ˙$C_2H_3$. The Cu (111)-F-far shows a reaction barrier of 0.12 eV at 0 V vs. RHE, which is reduced to 0.09 eV at −1.0 V vs. RHE (Supplementary Fig. S42), demonstrating the promotion of electrochemical steps by adjusting the applied potential. Moreover, the $C_2H_2$ semi-hydronation barrier of Cu–Cu (111)-F-far is lower than both the $C_2H_2$ semi-hydronation barrier of Cu (111) (0.21 eV), and the $H_2$ generation barrier of itself (0.29 eV, Supplementary Fig. S43). Thus, the HER process is suppressed, and the overall ESAE process is promoted over Cu–F.

### Direct electroreduction of acetylene to polymer-grade ethylene

Motivated by the impressive FE for $C_2H_4$ and high current density exhibited by Cu–F, this study explores the direct conversion of $C_2H_2$ (70 mol% in Ar) to polymer-grade $C_2H_4$ using Cu–F. It is essential to consider the effect of low $C_2H_2$ partial pressure and the $C_2H_4$-rich environment on the ESAE process, as the $C_2H_2$ impurities in polymer-grade $C_2H_4$ must be below 5 ppm[35]. Supplementary Fig. S44 demonstrates that the current density and FE for $C_2H_4$ decrease with the reduction in the $C_2H_2$ concentration, particularly at a low concentration of 1 mol%. Despite this, when ESAE is performed using a 25 cm² flow cell (electrode area: 25 cm²) with 1 mol% $C_2H_2$ in $C_2H_4$ (Supplementary Fig. S45), remarkable $C_2H_2$ conversion and $C_2H_4$ selectivity are still achieved (Fig. 4a and Supplementary Figs. S46 and S47). Furthermore, the residual $C_2H_2$ in the outlet gas remains below 5 ppm for up to 13.5 h at an applied cell voltage ($E_{cell}$) of −2.2 V (Fig. 4b). This demonstrates the efficient ESAE capability of Cu–F, even in a $C_2H_4$-rich environment with a low $C_2H_2$ partial pressure.

The direct electroreduction of acetylene to polymer-grade ethylene is performed using a custom-made tandem device consisting of a 1-cm² flow cell and a 25-cm² flow cell (Supplementary Fig. S48). The design is based on the capacity of the 1-cm² flow cell to deal with high-concentration $C_2H_2$ feed gas, and the ability of the 25-cm² flow cell for converting residual $C_2H_2$ at low concentration. Initially, $C_2H_2$ (70 mol% in Ar) is pumped into the 1-cm² flow cell, and the outlet gas is then immediately introduced into the 25-cm² flow cell without any purification (Fig. 4c). The optimal flow rate is determined to be 6 mL min⁻¹, as it achieves a high single-pass $C_2H_2$ conversion of approximately 80%

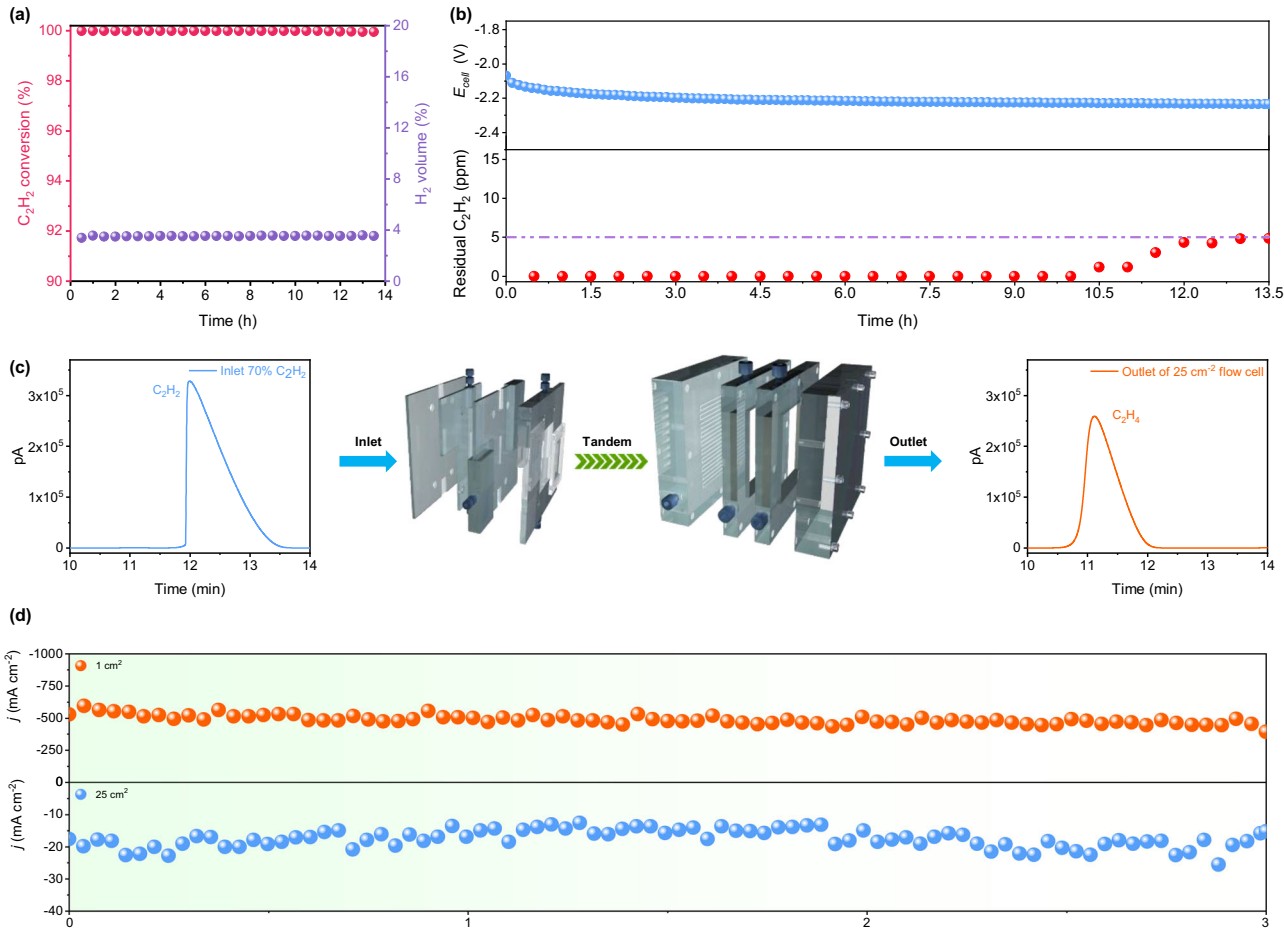

**Fig. 4 | Electrosynthesis of polymer-grade for acetylene (70% in Ar). a** $C_2H_2$ conversion and **b** long-term operation of ESAE over Cu−F under ethylene-rich environment, measured using a three-electrode flow cell (25 cm$^2$) in 1 M KOH at room temperature under 1 mol% $C_2H_2/C_2H_4$ flow (20 mL min$^{-1}$), constant current density set as 40 mA cm$^{-2}$. **c** Schematic of the tandem flow cells and the chromatograms at different locations of the gas line. **d** Long-term operation of ESAE in the tandem flow cells, 70 mol% $C_2H_2$ in Ar, flow rate 6 mL min$^{-1}$, constant potentials set as −1.1 and −0.6 V (vs. RHE) for 1-cm$^2$ and 25-cm$^2$ flow cells, respectively.

and $C_2H_4$ selectivity over 90% in the 1 cm$^2$ flow cell (Supplementary Figs. S18 and S49). As shown in Fig. 4d, the current density over Cu−F reaches 500 mA cm$^{-2}$ in the 1 cm$^2$ flow cell and 20 mA cm$^{-2}$ in the 25 cm$^2$ flow cell, and maintaining stable for up to 3 h without significant decay. Furthermore, minimal residual $C_2H_2$ is detected in the final $C_2H_4$ production, and the carbon loss in the tandem device is negligible (Fig. 4c and Supplementary Fig. S50). The high current density, robust stability, and high $C_2H_4$ selectivity of Cu−F in the tandem device highlight its potential application for the direct generation of polymer-grade ethylene from acetylene on an industrial scale.

In summary, halogen-doped Cu catalysts have been prepared as efficient electrocatalysts for the semihydrogenation of acetylene to ethylene. The optimized Cu−F catalyst, featuring F atoms adsorbed on the Cu (111) plane, exhibits outstanding performance in acetylene electroreduction. It achieves a current density of 1 A cm$^{-2}$ at a potential of −1.5 V (vs. RHE), and the Faraday efficiency for ethylene exceeds 90% over a wide potential range from −0.5 to −1.3 V (vs. RHE). In addition, the single-pass acetylene conversion varies between 30% and 80% depending on the flow rate. In situ spectroscopy, in combination with DFT calculations, reveals that the presence of F atoms enhances the water dissociation ability of adjacent Cu sites. This promotes the generation of active hydrogen species that are subsequently transferred to the adsorbed acetylene, thereby boosting the ESAE pathway. Furthermore, the Cu−F catalyst demonstrates almost complete conversion of acetylene (70 mol% in Ar) to ethylene in a tandem flow cell

device, highlighting its significant potential for direct electroreduction of acetylene into polymer-grade ethylene. This study opens new opportunities for the rational design of catalysts for large-scale electrosynthesis of ethylene and other essential industrial raw materials.

## Methods

### Chemicals and materials

All reactants are analytical grade without purification as them received. Ammonium bifluoride ($NH_4HF_2$), copper nitrate trihydrate ($Cu(NO_3)_2 \cdot 3H_2O$), ammonium chloride ($NH_4Cl$), ammonium bromide ($NH_4Br$), ammonium iodide ($NH_4I$), tert-butylalcohol (*t*-BuOH), Tetra-methylammonium hydroxide solution (TMA$^+$OH$^-$) were purchased from Beijing InnoChem Science&Technology Co., Ltd. Ethanol ($C_2H_6OH$), potassium hydroxide (KOH), sodium hydroxide (NaOH), N,N-dimethylformamide (DMF), 25% ammonia solution ($NH_3.H_2O$) were purchased from Sinopharm Chemical Company. Nafion® D-521 dispersion was purchased from Alfa Aesar. The GDL (SGL, 39BB), nickel foam (1.0-mm thickness), anion exchange membrane (fumasep® FAA-3-50) were purchased from Fuel Cell Store. The ultrapure water involved in experiments was prepared with a resistivity of 18.2 MΩ.

### Preparation of Cu(OH)F

Cu(OH)F was prepared according to previous report[29]. Typically, 114.1 mg (2 mmol) $NH_4HF_2$ was dispersed in 50 mL DMF with vigorous stirring at room temperature for at least 40 min, then the solution correspondingly

changed to blue after 483.2 mg (2.0 mmol) Cu(NO$_3$)$_2$·3H$_2$O was introduced and stir for 30 min. The homogeneous liquid was poured into 100 mL hydrothermal reactor and sealed, then heated at 160 °C for 4 h. The green powder was collected by employing centrifuge process with ethanol and water washing in turn three times after it cooled down. After dried at 60 °C in vacuum, Cu(OH)F is obtained.

## Preparation of Cu(OH)X (X = Cl, Br, I)

Cu$_2$(OH)$_3$Cl, Cu$_2$(OH)$_3$Br, and CuI were fabricated also as previously reports with some modifications[29]. Generally, 300 mg NH$_4$Cl, NH$_4$Br, and NH$_4$I were respectively added in three-neck flask with a reflux condenser followed by placed 50 mg Cu(OH)F. Then 50 mL C$_2$H$_5$OH/H$_2$O (v/v, 49:1) was filled to form suspension liquid. After stir 1 h, the mixture was heated to 80 °C for 36 h. Through similar procedure of purification mentioned above, the concentrated precursor finally obtained by dried in vacuum at 60 °C.

## Preparation of Cu(OH)$_2$ precursor

Cu(OH)$_2$ precursor could be prepared through a typical method[36]. In total, 1.3 g Cu(NO$_3$)$_2$·3H$_2$O was dissolved in 100 mL water, then 30 mL 0.15 M NH$_3$.H$_2$O was also introduced in above Cu$^{2+}$ solution. Subsequently, Cu(OH)$_2$ precipitate was presented after 10 mL 1 M NaOH solution added in it with stirring for 30 min. Through filtration and vacuum drying, blue solid product was harvested.

## Preparation of Cu−F gas diffusion electrode

The working electrode was prepared through in situ electro-derivation of the relevant precursor on GDL surrounding 1 M KOH. Here, take the process of fabricating Cu−F GDE as an example. 5 mg Cu(OH)F precursor was dispersed in 0.75 mL C$_2$H$_5$OH/H$_2$O (v/v, 1:1) with 25 μl nifion binder, then uniform suspension ink was formed by ultrasonic dispersion method within 1 h. Afterward, all ink spray onto 0.5 × 2.0 cm GDL on the top of a heating plat at 65 °C to evaporate solvent with mas loading controlled at 1.0–1.2 mg. For 5 × 5 cm GDE, Cu(OH)F mass loading was about 25.0–30.1 mg. GDL with loading Cu(OH)F precursor as working electrode was in situ reduced in a flow cell and was immersed in Ar (30 mL min$^{-1}$) and 1 M KOH at 1.6 V (vs. Ag/AgCl) for 300 s. The obtained catalyst was labeled as Cu−F.

## Preparation of Cu-X (X = Cl, Br, I) and Cu NP gas diffusion electrode

Cu-X GDEs were prepared as the same procedures as those of Cu−F. The GDL was loaded Cu$_2$(OH)$_3$Cl, Cu$_2$(OH)$_3$Br, CuI, and Cu(OH)$_2$ precursor, respectively. Then the catalysts were marked as Cu−Cl, Cu−Br, Cu−I, Cu NP after in situ electroreduction.

## Electrochemical measurements

CHI 1130c and 1120c were employed and coupled with a typical flow cell consisted of gas chamber, cathodic chamber, and anodic chamber in all experiments. Generally, GDL with catalyst as a working electrode, nickel foam as a counter electrode, both chambers were separated by an anion exchange membrane, and Ag/AgCl electrode as a reference electrode constituted electrocatalytic system. Ar and 70% C$_2$H$_2$ gas rate were set as 30 mL min$^{-1}$, otherwise mentioned in this work and electrolyte flow rate was set at 1 mL min$^{-1}$. The single-cell ESAE experiments were conducted at different potential utilizing i–t curve, then the obtained gas product was directly provided access to gas chromatography to quantitatively analyze component and liquid product was analyzed via $^1$H NMR spectroscopy by mixing 500 μl sample with 200 μl D$_2$O and 0.1 μL DMSO. All potentials of LSV curvy were converted to the RHE scale according to Eq. (1) without solution resistance compensation.

$$E_{RHE} = E_{Ag/AgCl} + 0.197 + 0.0592 \times pH \qquad (1)$$

For a three-electrode flow cell tandem system, mainly consisted of fore-cell (1 cm$^2$) and post-cell (25 cm$^2$), 70% C$_2$H$_2$ feed gas flew into the fore-cell then immediately transformed into C$_2$H$_4$ and C$_2$H$_2$ mixed gas at 500 mA cm$^{-2}$ and 6 mL min$^{-1}$. Afterward, outlet gas of fore-cell was served as feed gas of post-cell which the connecting line between them was kept to the minimum to shorter the dead volume. Residual C$_2$H$_2$ could be completely converted to C$_2$H$_4$ at 20 mA cm$^{-2}$ and 6 mL min$^{-1}$ in post-cell. Individual pumps provided 1 M KOH electrolyte to tandem system as fore-cell at 1 mL min$^{-1}$ and post-cell at 5 mL min$^{-1}$. The whole process carbon balance reached 97–99%.

For a two-electrode flow cell system, assembled as above with larger geometric area, chronopotentiometry was introduced to purify ethylene through 25 cm$^2$ Cu−F GDE at 20 mL min$^{-1}$ and 40 mA under ethylene-rich feed gas (1% C$_2$H$_2$, 20% C$_2$H$_4$, Ar compensation). Electrolyte flow rate set at 5 mL min$^{-1}$. The R$_u$ resistances at working conditions in electrode system are listed in the Supplementary Table S4.

All the electrochemical performances are presented without the IR compensation, except the stability test in the 1 cm$^2$ flow cell (Fig. 2e). As the flow cell (1 cm$^2$) stability test, the curvy was compensated at 200 mA with the solution resistance was about 9.4 Ω.

## Performance assessment

Gas products (C$_2$H$_4$, C$_2$H$_2$, C$_4$, H$_2$) were analyzed through gas chromatography (Panna, A6) coupled with a FID detector and a TCD detector. The Plot Al$_2$O$_3$ column separated ethylene, acetylene and C$_4$, while Porapak Q and Molecular Sieve 5 A columns separated H$_2$. An external standard method was applied to estimate the concentration of component of gas products. The partial current density of products was calculated as Eq. (2)[37]

$$j_{\text{partial}} = \frac{Q_{gas}}{t} = \frac{v \times t \times \delta / V_m \times n \times F}{t} = \frac{v \times \delta \times n \times F}{V_m} \times 100\% \qquad (2)$$

Where, $v$: the flow rate of feed gas, mL s$^{-1}$,
   $\delta$: gas product concentration calculated by calibration curve,
   $n$: the number of electrons transferred of species,
   $F$: Faraday's constant, 96,485 C mol$^{-1}$,
   $V_m$: 24 L mol$^{-1}$.

Faraday efficiency was calculated as Eq. (3)[37]

$$FE(\%) = \frac{Q_{gas}}{Q_{total}} = \frac{j_{partial} \times t}{J_{total} \times t} = \frac{j_{partial}}{J_{total}} \times 100\% \qquad (3)$$

Where, $J_{total}$: the total current density,
   $Q_{total}$: the charge number.

The formation rate of every component was based on Eq. (4)[29]

$$R = \frac{Q_{total} \times FE}{F \times n \times t \times S} \qquad (4)$$

Where, $t$: the electrolysis time (h) corresponding to $Q_{total}$,
   $S$: the geometric area of the working electrode (cm$^2$).

The calculation of C$_2$H$_2$ conversion and C$_2$H$_4$ selectivity was premise on carbon balance and according to Eqs. (5)[22] and (6)[19]

$$C(\%) = \frac{c_{feed} - c'}{c_{feed}} \times 100\% \qquad (5)$$

$$S(\%) = \frac{c_{feed} - c'}{c_{feed} - c' + c_{C_2H_6} + 2 \times c_{c_4}} \times 100\% \qquad (6)$$

Where, $c_{feed}$: concentration of C$_2$H$_2$ feed gas,
   $c'$, $c_{C_2H_6}$, $c_{c_4}$: the concentration of C$_2$H$_2$, C$_2$H$_6$, C$_4$ in gas products.

H$_2$ generated in ESAE process was estimated as Eq. (7)[19]

$$H_2(\%) = \frac{\upsilon_{out}}{\upsilon_{feed}} \times 100\% \qquad (7)$$

Where, $\upsilon_{out}$: the H$_2$ volume in gas products,

$\upsilon_{feed}$: the volume of feed gas.

Theoretical conversion current was premise on entirely C$_2$H$_2$ conversion based on Eq. (8)[19]

$$I = \frac{n \times F \times P \times \upsilon_x}{R \times T} \qquad (8)$$

Where, $P$: the atmospheric pressure, $101.3 \times 10^3$ Pa,

$R$: the molar gas constant, 8.314 J (mol K)$^{-1}$,

$T$: the temperature, 293.15 K,

$\upsilon_x$: the velocity of acetylene in mixed gas. As simulated feed gas (1% C$_2$H$_2$, 20% C$_2$H$_4$) at 20 mL min$^{-1}$, the $\upsilon_x$ could be calculated as $3.3 \times 10^{-6}$ S$^{-1}$ and theoretical conversion current was evaluated as 26.7 mA. When the velocity of feed gas came to 10, 30, 40, and 50 mL min$^{-1}$, $\upsilon_x$ was calculated as 13.4, 40.1, 53.4, and 66.8 mA, respectively.

## Characterizations

The X-ray diffraction (XRD) patterns were performed on Bruker D8 advance diffractometer with Cu Kα radiation. X-ray photoelectron spectra (XPS) were performed on a ESCALAB Xi$^+$ photoelectron spectrometer with monochromatic Al Kα X-rays to verify the valence state of Cu and halogen. Casa XPS was introduced to analyze spectra calibrated by the C 1 s spectrum (284.8 eV). Scanning electron microscopy (SEM) was harvested by Apreo S instrument to reveal morphology of catalysts. Transmission electron microscopy (TEM), high-resolution TEM (HRTEM) images, the corresponding energy-dispersive X-ray spectroscopy (EDS), and selected area electron diffraction (SAED) were measured on a FEI Talos F200X. Aberration-corrected HAADF-STEM images were performed on a JEOL JEMARM200F TEM/STEM system. The X-ray absorption fine structure spectra were collected at the Beijing Synchrotron Radiation Facility (BSRF) in China. The acquired EXAFS data were extracted and processed according to the standard procedures using the ATHENA module implemented in the IFEFFIT software packages, and copper foil and copper oxide were used as references to identify elaborate valence and coordination environment of Cu in catalysts. Operando Raman spectra were measured in a three-electrode observable window electrochemical cell with a counter electrode of Pt wire and Ag/AgCl under controlled potentials in 1 M KOH electrolyte, and a controlled active area of 0.384 cm$^2$ by an insulation layer on carbon paper sprayed with 1 mg Cu(OH)F was used as the working electrode. Raman spectra were collected using a Raman spectrometer (Horiba labRAM HR Evolution) by a 532 nm laser focusing on the organic materials distributed on sample surface after precursor in situ derivation under Ar then switched to C$_2$H$_2$.

## Computational details

In DFT calculations, a $2 \times 2$ supercell of the Cu (111) slab model was constructed. Subsequently, a fluorine (F) atom was introduced onto the hollow site of the Cu (111) surface, resulting in the formation of Cu (111)-F. To prevent interactions between images, a vacuum layer with a thickness of 15 Å along the $z$-direction was implemented. Structural optimization calculations were practiced via the Vienna Ab-initio Simulation Package (VASP) with the projector augmented wave (PAW) method. The Perdew–Burke–Ernzerhof (PBE) functional, in conjunction with the DFT-D3 correction, was employed to handle the exchange function. The plane-wave basis cut-off energy was set at 450 eV. For geometry and lattice size optimization, Brillouin zone integration

utilized a Gamma k-point mesh of $3 \times 3 \times 1$. Self-consistent calculations adhered to a convergence energy threshold of $10^{-5}$ eV. Equilibrium geometries and lattice constants were optimized, with a maximum stress on each atom kept within 0.02 eV Å$^{-1}$.

In the computation of Gibbs free energy, the hydrogen adsorption model was constructed using the computational hydrogen electrode (CHE) model. The C$_2$H$_2$ hydrogenation steps to C$_2$H$_4$ were delineated as follows:

$* + C_2H_2 \rightarrow *C_2H_2$

$*C_2H_2 + H^+ + e^- \rightarrow *C_2H_3$

$*C_2H_3 + H^+ + e^- \rightarrow *C_2H_4$

$*C_2H_4 \rightarrow * + C_2H_4$

The hydrogen combination proceeded through these steps:

$* + H^+ + e^- \rightarrow *H$

$*H + H^+ + e^- \rightarrow * + H_2$

While the water dissociation process involved the following steps:

$* + H_2O \rightarrow *H_2O$

$*H_2O \rightarrow *H{-}OH$

$*H{-}OH \rightarrow *H + OH^-$

The Gibbs free energy of the H atom was computed based on $H_2 \rightarrow H^+ + e^-$, where $G(H^+) = 1/2 \, G(H_2)$. Entropies of free molecules H$_2$ and H$_2$O were referenced to the NIST database, while those of free molecules C$_2$H$_2$ and C$_2$H$_4$ were obtained from the vaspkit interface. The Gibbs free energy of intermediates was calculated as G = E + E$_{zpe}$ - TS, where E, E$_{zpe}$, and S represent the energy, zero-point energy, and entropy of surface adsorbing intermediates, respectively. In addition, the Kelvin temperature T was set at 298.15 K, with both E$_{zpe}$ and TS acquired through the vaspkit interface. The pH value of 14 was set to simulate the reaction conditions.

## Data availability

The data that support the findings of this study are available within the article and its Supplementary Information files. All other relevant data supporting the findings of this study are available from the corresponding authors upon reasonable request. Source data are provided with this paper.

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

## Acknowledgements

The authors thank the photoemission endstation beamline 1W1B station in the Beijing Synchrotron Radiation Facility (BSRF) for help with the characterizations. This work is financially supported by the National Natural Science Foundation of China (22275147 to Y.Q., 52301289 to K.L., and 21902150 to Y.Q.), Natural Science Basic Research Program of Shaanxi (2022JQ-082 to K.L. and 2022JM-018 to Y.W.).

## Author contributions

Y.Q. and K.L. designed the experiments and performed DFT calculation. L.B., Y. Wang, and Z.H. conducted the experiments and performed the catalysts. L.Z. performed X-ray absorption fine structure spectra measurements. All authors contributed to the scientific interpretation and we sincerely thank J.B. and Y. Wu for profound suggestions and insightful guidance.

## Competing interests

The authors declare no competing interests.
