## [Peer Review File · Nature Communications]

REVIEWER COMMENTS

Reviewer #1 (Remarks to the Author):

In the manuscript 'Industrial-Level Electrocatalytic Acetylene to Polymer-Grade Ethylene through Hydrogen-Localization Transfer over Fluorine-Modified Copper', Bai et al. report on the synthesis of copper-based electrodes for the selective electrochemical reduction of acetylene to ethylene. The authors propose the formation of a partially fluorinated (halogenated) Cu electrode as active site with higher intrinsic activity for the hydrogenation of acetylene over the formation of ethane and the competing HER.

The authors reduce Cu(OH)F electrochemically at potentials about 1 V negative of the $\text{Cu}^{2+} \rightarrow \text{Cu}^0$ couple reported in typical Pourbaix diagrams. Have the authors ever measured CVs on the Cu(OH)F electrode to evaluate if there are notable redox features that could evince said $\text{Cu}^{2+} \rightarrow \text{Cu}^+$, $\text{Cu}^+ \rightarrow \text{Cu}^0$, or $\text{Cu}^{2+} \rightarrow \text{Cu}^0$ features? Upon exposure to air and/or open circuit conditions, it is inevitable that a fraction of predominantly surface Cu⁰ sites would reoxidize again to Cu⁺ or Cu²⁺. It remains therefore unclear how much information ex situ studies provide on the nature of the active site.

Similarly, electrochemical hydrogenation reactions are usually done at potentials far negative of the $\text{Cu}^+ \rightarrow \text{Cu}^0$ redox couple – the natural assumption would be that deeply reduced Cu⁰ might be the active site.

It appears that the electrochemical performance of Cu nanoparticle and the related Cu halide samples are rather similar. Could their differences be attributed to differences in morphology (e.g. ECSA)? Would a much easier explanation be the formation of different Cu⁰ surfaces in the presence of halides with therefore different densities of Cu⁰ active sites (planar, edges, etc.) that all have different intrinsic rates in HER and C₂H₂/C₂H₄ hydrogenation?

The authors rationalize the different intrinsic activity for acetylene conversion to the interaction between solvated cation and F (it is not clear if they mean solvated fluoride in solution or surface-adsorbed fluorine). The mechanism through which cations promote electrochemical hydrogenations is not clear. It appears that the difference in electrochemical performance follows the same trend as the cation diffusion coefficient, we can ergo argue that the transference number and thus interfacial activity of cations is largely different. This would change this effect to a transport-related phenomenon which would be augmented by different electrode morphologies (CuF and Cu might therefore behave somewhat differently).

How is the electrostatic potential accounted for in the DFT simulations? It currently appears that no electrostatic potential is considered in their atomistic simulations which makes the correlation with electrolysis experiments doubt-worthy.

While it might be acceptable to not account for the cell resistance in the electrochemical experiments, reporting R_u values under different experimental conditions would help the reader judge how much E-dependence should be due to changes in resistance (since the authors pass a lot of current).

Claims of industrial level electrolysis are usually made based on no defined metrics. The authors do not specify what industrial level electrolysis conditions are for the electrochemical reduction of acetylene to ethylene and hence cannot claim to fulfill those. High current density is by no means a sufficient criterion to support this notion.

On some occasions, the authors qualitatively describe (and indeed judge) that e.g. CuF is much better than Cu for instance with regards to the H₂ FE. Instead of making this claim, the authors should rather quote numbers here and let the readers judge for themselves if this should be considered a significant difference.

On many occasions, figure captions are rather imprecise, reaction conditions for any of the electrochemical reaction in Fig. 3, Fig. 2a, b, c, d, and e (as well as all the associated supplementary figures) are unclear.

The authors do report an interesting approach to the electrochemical selective hydrogenation of acetylene but a lot of evidence on the active site structure and proposed mechanism is not very convincing. Overall, I think, however, that this manuscript could become publishable after careful revision.

Reviewer #2 (Remarks to the Author):

The manuscript authored by Bai et al. presents a significant advancement in the domain of C₂H₄ production derived via electrocatalytic semi-hydrogenation of C₂H₂. The researchers drew inspiration from the CO₂RR field and devised a Cu-F/GDE electrode. This electrode achieved a substantial C₂H₂ to C₂H₄ conversion and high Faradaic efficiency. The efficacy of Cu-F was attributed to the enhanced water dissociation rate facilitated by F⁻. It is suggested that the H^{*} was produced from Cu sites near F and spillover onto Cu sites far from F, where the hydrogenation of C₂H₂ takes place. The work showcased in the manuscript is intriguing as the authors conducted a comprehensive investigation from various standpoints encompassing catalyst design, device engineering, process optimization, and mechanistic elucidation through DFT and in situ Raman spectroscopy. Given its breadth and depth, this work appears suitable for publication in Nature Communications. Nevertheless, I would like to offer some comments and pose questions that could potentially enhance the manuscript's quality.

1. There is a typo on Page 4, Line 103: It should be NH₄HF₂ instead of NH₄HF.

2. For figures lacking a legend, it would be beneficial to provide additional contextual information within the caption. For instance, in Fig 2a, clarification regarding the conditions of the dashed and solid lines would be helpful.

3. The size of Figure 2b is insufficient. To improve visibility, relocating the inset image to the Supplementary Information and labeling the shift in onset potentials directly onto the plot could be considered.

4. Could you please elaborate on the rationale behind reporting the system without iR compensation? Did the solution resistance exhibit significant variability across runs or with differing feed concentrations?

5. Has the I-V curve been observed to alter when the gas flow rate is decreased from 30 to 6 mL/min?

6. The section pertaining to mechanistic insights is captivating. However, the descriptions of the key experiments appear rather concise and abrupt. It might be advantageous to expound upon this section. To facilitate comprehension, the following elaborations could be valuable:

- Could you elucidate how the insensitivity of Faradaic efficiency to pH alterations excludes the plausibility of a PCET process, especially when the proton source is H₂O instead of H⁺?

- Can you detail how tert-butanol functions as an H atom capturer? Were different reaction products observed upon the addition of tert-butanol?

- Regarding the KIE experiment, could you offer further clarification on how high KIE values indicate that the H₂O dissociation is the rate-determining step? Hence, lower KIE values could signify improved water dissociation.

- The authors propose that H₂O dissociation transpires at the Cu-F near site, with H^{*} subsequently transferring to the Cu-F far site. Could you provide insight into the distance between individual F atoms? Additionally, have energy barriers for the spillover process been investigated?

- The distinct in situ Raman peaks corresponding to C=C on Cu-F and Cu NP raise questions. How did the authors verify that the peak at 1554 cm⁻¹ indeed represents C=C? Furthermore, the doublet peaks between 1200-1600 appear akin to typical I_d and I_g peaks of carbon substrates. Could you shed light on their differentiation?

7. Kindly elucidate the rationale behind designing the tandem 1-25 cm² device in such a configuration. What considerations guided this particular process design?

8. The demonstration of stability over a mere 3000s appears rather brief. What intrinsic limitations influence this choice of duration?

Responses to Reviewers:

The reviewers' comments are laid out below in *Italic font* and specific concerns have been numbered. Our responses are given in normal font and changes/additions to the manuscript and supplementary information are highlighted by using **red colored text**.

Following is our point-to-point answer to all the comments:

Comments from Reviewer #1

In the manuscript 'Industrial-Level Electrocatalytic Acetylene to Polymer-Grade Ethylene through Hydrogen-Localization Transfer over Fluorine-Modified Copper', Bai et al. report on the synthesis of copper-based electrodes for the selective electrochemical reduction of acetylene to ethylene. The authors propose the formation of a partially fluorinated (halogenated) Cu electrode as active site with higher intrinsic activity for the hydrogenation of acetylene over the formation of ethane and the competing HER.

1. *The authors reduce Cu(OH)F electrochemically at potentials about 1 V negative of the $\text{Cu}^{2+} \rightarrow \text{Cu}^0$ couple reported in typical Pourbaix diagrams. Have the authors ever measured CVs on the Cu(OH)F electrode to evaluate if there are notable redox features that could evince said $\text{Cu}^{2+} \rightarrow \text{Cu}^+$, $\text{Cu}^+ \rightarrow \text{Cu}^0$, or $\text{Cu}^{2+} \rightarrow \text{Cu}^0$ features? Upon exposure to air and/or open circuit conditions, it is inevitable that a fraction of predominantly surface Cu^0 sites would reoxidize again to Cu^+ or Cu^{2+} . It remains therefore unclear how much information ex-situ studies provide on the nature of the active site.*

Answer: Thanks to the comment. In this work, the Cu(OH)F was reduced at 0.6 V vs. RHE (1.6 V vs. Ag/AgCl). To clarify the electrical reduction process, the LSV results (from 0 V to -1 V vs RHE) had been recorded in 1 M KOH solution for Cu(OH)F. As shown in Figure S6, the LSV curve illustrates a reduction peak at -0.35 (V vs. RHE), demonstrating the occurrence of the electroreduction of Cu^{2+} to Cu^0 . Our results in line with the reported electroreduction process on Cu-based materials (Nature Sustain., 2023, 6, 827).

We agreed to the reviewer that the Cu^0 would inevitably be oxidated after peeled from the GDL for ex-situ investigation. In order to make the nature of the active site clear, an in-situ XAFS measurement had been conducted during the electroreduction of Cu(OH)F for forming Cu-F catalyst, and during the subsequent acetylene semi-hydrogenation. The conditions were set as the same with the fabricating of Cu-F from Cu(OH)F, that was -0.6 V vs. RHE (-1.6 V vs. Ag/AgCl) under Ar flow.

Figure S30 shows the in-situ XANES spectra, although a significantly decreased oxidation state of Cu was observed after reducing Cu(OH)F under Ar flow for 5 min, a partial positive oxidation state between 0 to +1 was still indicated by the intensity of the absorption edge. The FT EXAFS spectra further indicated the co-existence of F-Cu and Cu-Cu coordination after the sample was reduced for 5 min in Ar flow. These results demonstrated the innate Cu⁺ sites in the fresh Cu-F, which stemmed from the partial F-Cu coordination. Moreover, after the catalyst was reduced for 5 min, we switched the Ar flow to the C₂H₂ flow to conduct a 120 min acetylene semi-hydrogenation process, after that the in-situ XAFS was taken. Interestingly, the oxidative state of Cu still located between 0 to +1 with slight decreasing, and the F-Cu coordination was maintained also. Obviously, although a C₂H₂-hydrogenation potential that far negative of the Cu⁺ to Cu⁰ redox couple was used, the deep reduction of Cu^{²+} to Cu⁰ was absent. Therefore, the in-situ XAFS results indicated the intrinsic Cu⁺ sites in the as prepared and used Cu-F catalysts.

The related results had been added into the Supporting Information files as Figure S6 and S30, and discussed in the revised manuscript.

Page 5, line 120-121: “The LSV curve of Cu(OH)F reduction shows a peak at -0.35 V vs. RHE (Figure S6), indicating the occurrence of the electroreduction on Cu²⁺ during the Cu-F generation.”

Page 8, line 186-192: “The nature of the active sites in Cu-F during the ESAE process is investigated by in-situ XAFS (Figure S30). The freshly generated Cu-F (5 min in Ar) shows an average oxidation state of Cu between 0 to +1 with Cu-Cu and F-Cu coordination, confirming the innate Cu⁺ sites accompany Cu⁰ sites. Moreover, after triggering the ESAE by switch the gas flow to C₂H₂ for 120 min, the average oxidation state and coordination environment of Cu show slight changes, suggesting the maintaining of the F-Cu interaction.

Figure S6. The LSV curve of the Cu(OH)F reduction recorded in 1 M KOH with a Ar flow (30 ml min⁻¹) and a scan rate of 10 mV s⁻¹.

Figure S30. The in-situ XAFS measured on the generation process of Cu-F and the subsequent acetylene semi-hydrogenation process. Measured at -0.6 V vs. RHE (-1.6 V vs. Ag/AgCl).

2. Similarly, electrochemical hydrogenation reactions are usually done at potentials far negative of the $\text{Cu}^+ \rightarrow \text{Cu}^0$ redox couple – the natural assumption would be that deeply reduced Cu^0 might be the active site.

Answer: Thank you for the enlightening comments. Based on the answer to comment 1, the in-situ XAFS result (Figure S30) had confirmed the maintain of the Cu^+ sites during the electrochemical hydrogenation reactions, even at a potential that negative of the Cu^+ to Cu^0 redox couple. This can be attributed to the competitive relation between the electroreduction of C_2H_2 and the electroreduction of $\text{Cu}^{\&+}$. As indicated in Figure S6, the reduction of $\text{Cu}^{\&+}$ started at -0.23 V vs. RHE, however the onset potential for C_2H_2 semi-hydrogenation was just -0.12 V vs. RHE (Figure S21), confirming the electroreduction of C_2H_2 is much easier to happen than the electroreduction of $\text{Cu}^{\&+}$. Therefore, the Cu^+ sites in Cu-F were protected from being deeply reduced by the competitive advantage of C_2H_2 semi-hydrogenation in the reaction process. Analogous results that $\text{Cu}^{\&+}$ can be preserved at relatively negative potential (<-0.6 V vs. RHE) had been reported based on previous works (Nature Catal., 2020, 3, 478; Nature Catal., 2021, 4, 565). These related results had been discussed at page 8, line 192-195:

“Considering the much more positive onset potential of ESAE (-0.12 V, Figure S21) than the reduction potential of $\text{Cu}^{\&+}$ (-0.23 V, Figure S6), C_2H_2 is much easier to be reduced than $\text{Cu}^{\&+}$. Therefore, the F-Cu coordination is protected from being deeply reduced, which contributes to the superior EASE performance of Cu-F.”

Figure S30. The in-situ XAFS measured on the generation process of Cu-F and the subsequent acetylene semi-hydrogenation process. Measured at -0.6 V vs. RHE (-1.6 V vs. Ag/AgCl).

Figure S6. The CV curve on the Cu(OH)F recorded in 1 M KOH with a Ar flow (30 ml min^{-1}) and a scan rate of 10 mV s^{-1} .

Figure S21. The onset potentials over Cu-F for C_2H_4 and H_2 generation. Measured using a three-electrode flow cell (1 cm^2) in 1 M KOH at room temperature under 70 mol% $\text{C}_2\text{H}_2/\text{Ar}$ flow (30 ml min^{-1}). The results are presented without iR compensation.

3. It appears that the electrochemical performance of Cu nanoparticle and the related Cu halide samples are rather similar. Could their differences be attributed to differences in morphology (e.g. ECSA)? Would a much easier explanation be the formation of different Cu^0 surfaces in the presence of halides with therefore different densities of Cu^0 active sites (planar, edges, etc.) that all have different intrinsic rates in HER and C_2H_2/C_2H_4 hydrogenation?

Answer: Thanks to your professional comments. To rule out the effect of the morphology and obtain a clearer idea of the intrinsic activity, we measured the ECSA for all halide doped Cu and Cu NP. As seen in Figure S16-17, the ECSA-normalized current densities follow the same trend with the apparent EASE performance in Figure 2a, and the Cu-F show best result. These results demonstrate the superior intrinsic activity of Cu-F, and exclude the effect of the potential difference in morphology. The ECSA results had been added into the revised manuscript as Figure S6-17, Table S2 and discussed at page 7, line 160-162:

“The ECSA-normalized current densities follow the same trend with the apparent EASE performance (Figure S16-17, Table S2), ruling out the effect of the morphology, and demonstrating the superior intrinsic activity of Cu-F.”

Figure S16. Electrochemical capacitance measurements. Cyclic voltammogram curves of (a) Cu NP, (b) Cu-F, (c) Cu-Cl, (d) Cu-Br and (e) Cu-I at different scan rate. (f) The corresponding charging current densities vs. applied scan rate.

Figure S17. ECSA-normalized electrocatalytic performance of various catalysts. (a) LSV curves of HER measured under Ar flow. (b) C₂H₄ formation rate at -1.0 V measured under 70 mol% C₂H₂/Ar flow. Measured using a three-electrode flow cell (1 cm²) in 1 M KOH at room temperature with gas flow rate of 30 ml min⁻¹. The results are presented without iR compensation.

Figure 2a. LSV curves in Ar (dash) and 70% C₂H₂/Ar (sphere).

Table S2. Capacitance, surface roughness factors and electrochemical surface areas (ECSA) for Cu-X (F, Cl, Br, I) and Cu NP

Catalyst	C_{dl} (mF cm ⁻²)	C_s^a (μF cm ⁻²)	ECSA (cm ²)
Cu-F	2.5		86.2
Cu-Cl	2.7		93.1
Cu-Br	2.8	29	96.6
Cu-I	2.9		100
Cu NP	3.3		113.8

^a C_s the corresponding smooth polycrystalline Cu electrode.

4. The authors rationalize the different intrinsic activity for acetylene conversion to the interaction between solvated cation and F (it is not clear if they mean solvated fluoride in solution or surface-adsorbed fluorine). The mechanism through which cations promote electrochemical hydrogenations is not clear. It appears that the difference in electrochemical performance follows the same trend as the cation diffusion coefficient, we can ergo argue that the transference number and thus interfacial activity of cations is largely different. This would change this effect to a transport-related phenomenon which would be augmented by different electrode morphologies (CuF and Cu might therefore behave somewhat differently).

Answer: Thanks to the comments. When discussing the interaction between the hydrated cation and F, we mean the surface F in Cu-F. However, we cannot deny that part of the F⁻ will leach into the electrolyte during the in-situ electrical generation of Cu-F from Cu(OH)F. To investigate the effect of the solvated fluoride in electrolyte, a contrast experiment was added. After the in-situ electrical generation of Cu-F, the used electrolyte was replaced by a brand new one that containing no F. As shown in Figure S29, the catalytic performance of Cu-F, including the LSV curves and Faraday efficiency almost have no change no matter the electrolyte is replaced or not, demonstrating the negligible effect of the solvated fluoride in electrolyte. In the revised manuscript, the additional experiment had been added as Figure S29 and discussed at page 7 line 180-184:

“It is inevitable that part of F will leach into the electrolyte during the in-situ electrical generation of Cu-F from Cu(OH)F. Figure S29 shows the EASE performance of Cu-F in a fresh electrolyte (F-free), the negligible changes on the activity exclude the effect of the solvated fluoride on the EASE process.”

The investigation on the effect of different hydrated cation in Figure 3d was used to confirm the role of the surface F on the water dissociation. (The hydrated cation means M^{&+}(H₂O)_n, where M^{&+} refers to the cation and n refer to the number of hydrations). The mechanism is that, during the electrocatalytic process the surface anion (e.g., F⁻, S^{&-}) interact with the hydrated cation in the Helmholtz layer via non-

covalent Coulomb interactions, contributing to the dissociation of H₂O to form active hydrogen species. Hydrated cation with larger n and radius (Na⁺(H₂O)₁₄ versus K⁺(H₂O)₇) can weaken this interaction, depress the dissociation of H₂O, and lead to a reduced catalytic performance. Therefore, if the surface F controls the water dissociation process, the Cu-F catalyst should show better performance with K⁺ than that with Na⁺ and TMA⁺, due to the smaller n and radius of the hydrated K⁺ (Nature Commun., 2022, 23, 5297; Nature Catal., 2020, 3, 478). As shown in Figure 3d, the catalytic performance of Cu-F, follow the trend of TMA⁺ < Na⁺ < K⁺, in line with the mechanism above, demonstrating the role of F in water dissociation.

Furthermore, as the reviewer points out, the diffusion indeed plays a significant role in numerous catalytic processes. However, in the case of the halogen-free catalyst Cu NP (Figure S38), the catalytic performance shows no obvious change no matter which cation is used, although these cations have different diffusion coefficients. Moreover, the answer to Comment 3 had excluded the effect of morphologies for Cu-F and Cu NP. Therefore, it can be concluded that the cation diffusion coefficient has restricted influence on the acetylene semi-hydrogenation process reported in this work, and the surface F plays a key role in regulating the water dissociation for efficient hydrogenation. Based on the reviewer's comment, the mechanism and the exposition about the hydrated cation investigation had been further clarified in the revised manuscript at page 9-10, line 239-249

“The hydrated cation would interact with the surface F in the Helmholtz layer, contributing to the dissociation of H₂O to form active hydrogen species, and this interaction is determined by the n and radius of the hydrated cation³⁴. As shown in Figure 3d and Figure S37, when replacing 1 M KOH by 1 M tetramethylammonium hydroxide (TMAH) or NaOH, both the C₂H₂ conversion and HER over Cu-F exhibit significant decay, which due to the weaker interaction of F with TMAH and Na⁺(H₂O)₁₃ than with K⁺(H₂O)₇, caused by their larger radius and n^{29, 34}. In contrast, no obvious change on C₂H₂ conversion and HER over Cu NP are observed, although cations with different nature (such as different diffusion coefficient) are used (Figure S38). The

investigation of hydrated cation reveals the key role of F doping for regulating water dissociation.”

Figure S29. The effect of the leached F in the electrolyte. (a) LSV curves. The orange sphere represents the curve measured in the original electrolyte used for generating Cu-F, which may contain the leached F; The blue line represents the electrolyte is replaced by a fresh one after the generation of Cu-F. (b) FE and the current density.

Figure 3d (left). C₂H₂ conversion in 70 mol% C₂H₂ in Ar with 1 M KOH, NaOH and TMAH electrolyte. **Figure S37 (right).** LSV curves in pure Ar over Cu-F with 1 M KOH, NaOH and TMAH electrolyte.

Figure S38. (a) C₂H₂ conversion in 70 mol% CH₂H₂ and (b) LSV in pure Ar over Cu NP with 1 M KOH, NaOH and TMAH electrolyte.

5. How is the electrostatic potential accounted for in the DFT simulations? It currently appears that no electrostatic potential is considered in their atomistic simulations which makes the correlation with electrolysis experiments doubt-worthy.

Answer: Thanks to the comments. In this work, the potential was set as 0 V vs. RHE for the DFT calculations. To further correlate the DFT and experimental investigations, an additional calculation for the free energy of C₂H₂ semi-hydration had been conducted at -1.0 V vs. RHE. As shown in Figure S42 and Figure 3i, the free energy diagram for various active sites at -1.0 V vs. RHE followed the same trend with that at 0 V vs. RHE, and the free energies of the electrochemical hydrogenation steps were reduced at the more negative potential. For instance, the barrier of the rate-determining step that *C₂H₂ hydrogenation to *C₂H₃ reduced from 0.12 eV (0 V vs. RHE) to 0.09 eV (-1.0 V vs. RHE), demonstrating the promotion of electrochemical steps by adjusting the applied potential. In the revised manuscript, the additional DFT results had been added as Figure S42 and discussed at page 10, line 267-271:

“As shown in Figure 3i, the rate determining step for the as illustrated C₂H₂ hydrogenation process on Cu sites is identified as the hydration of *C₂H₂ to *C₂H₃. The Cu (111)-F-far shows a reaction barrier of 0.12 eV at 0 V vs. RHE, which is reduced to 0.09 eV at -1.0 V vs. RHE (Figure S42), demonstrating the promotion of electrochemical steps by adjusting the applied potential.”

Figure 3i. Free energy diagram for the hydrogenation of C₂H₂ at 0 V vs. RHE.

Figure S42. Free energy diagram for the hydrogenation of C_2H_2 at $-1.0V$ vs. RHE.

6. While it might be acceptable to not account for the cell resistance in the electrochemical experiments, reporting R_u values under different experimental conditions would help the reader judge how much E -dependence should be due to changes in resistance (since the authors pass a lot of current).

Answer: Thanks to the comments. Based on the reviewer's suggestion, the R_u values had been provided in experimental section in SI files:

“The R_u resistances at working conditions in electrode system are listed in the table below:

Electrode system	Catalysts	Resistances (Ω)	Electrode area (cm^2)
Three electrode system	Cu-F	1.1	1
	Cu-Cl	1.2	
	Cu-Br	1.2	
	Cu-I	1.2	
	Cu NP	1.2	
Two electrode system	Cu-F	9.4	25
Three electrode system	Cu-F	1.5	
Two electrode system	Cu-F	9.7	
Three electrode system	Cu-F	1.1 @50 mol % C_2H_2	1
		1.1 @ 5 mol % C_2H_2	

All electrochemical performances are presented without the IR compensation, except the stability test in the 1 cm^2 flow cell (Figure 2e). As flow cell (1 cm^2) stability test,

the curvy was compensate at 200 mA with the solution resistance was about 9.4 Ω .”

7. Claims of industrial level electrolysis are usually made based on no defined metrics. The authors do not specify what industrial level electrolysis conditions are for the electrochemical reduction of acetylene to ethylene and hence cannot claim to fulfill those. High current density is by no means a sufficient criterion to support this notion.

Answer: Thank you for the professional comments. In this present work, the primary concern is to realize the C₂H₂ semi-hydration to C₂H₄ at a high current density, which is one of the crucial factors to face the potential industrial application. However, as pointed by the reviewer, the industrial level electrolysis cannot be represented by the high current density alone. Therefore, in the revised manuscript, we edited the title of this work to make it more reasonable:

“Efficient Industrial-Current-Density Acetylene to Polymer-Grade Ethylene via Hydrogen-Localization Transfer over Fluorine-Modified Copper”

8. On some occasions, the authors qualitatively describe (and indeed judge) that e.g. CuF is much better than Cu for instance with regards to the H₂ FE. Instead of making this claim, the authors should rather quote numbers here and let the readers judge for themselves if this should be considered a significant difference.

Answer: Thanks to the comments. In the revised manuscript, the comparison between different catalysts were supported by numbers, instead of the qualitative description. Such as:

Page 7, line 170-175: “The notable C₂H₄ selectivity of Cu-F in alkaline medium is revealed by the high Faraday efficiency (FE), which surpass 90% at a wide potential range from -0.5 to -1.3 V (vs. RHE, Figure 2c, Figure S22), better than the Cu NP (~80%) and other halogen doped Cu (Figure S23-26). In the same potential range, the highest H₂ FE over Cu-F is just 4.1% (-1.3 V), but for Cu NP it reaches 28.9% (-1.0 V).”

9. On many occasions, figure captions are rather imprecise, reaction conditions for any of the electrochemical reaction in Fig. 3, Fig. 2a, b, c, d, and e (as well as all the

associated supplementary figures) are unclear.

Answer: Thanks to the comments. In the revised manuscript, the electrochemical reaction condition had been described clearer in the figure captions, including Figure 2, Figure 3 and the associated supplementary figures:

“Figure 2. Electrocatalytic performance of Cu-F in 1 cm² flow cell. (a) LSV curves in Ar (dash) and 70% C₂H₂/Ar (sphere). (b) partial current density for C₂H₄ and H₂. (c) Faraday efficiency for C₂H₄ and H₂ at various potential and the corresponding current density. (d) C₂H₄ formation rate. (e) long-term stability at a constant current density of 200 mA cm⁻². All tests are measured using a three-electrode flow cell (1 cm²) in 1 M KOH at room temperature under 70 mol% C₂H₂/Ar flow (30 ml min⁻¹). The results are presented without iR compensation except e. For (e) the R=9.4 Ω.”

“Figure 3. Mechanistic insight. (a, b) Potential-dependent operando Raman spectra, measured using a three-electrode observable cell in 1 M KOH at room temperature under 70 mol% C₂H₂/Ar flow (30 ml min⁻¹). (c) Kinetic isotope effect, KIE. (d) C₂H₂ conversion under 70 mol% C₂H₂/Ar flow (30 ml min⁻¹) in 1 M KOH, NaOH and TMAH electrolyte. (e) Adsorption energy for C₂H₂ and H₂O, (f) free energy diagram for water dissociation and (g) adsorption energy for hydrogen atom, at 0 V vs. RHE. (h) Schematic illustration of the ESAE pathway over Cu-F. (i) Free energy diagram for the hydrogenation of C₂H₂ at 0 V vs. RHE.”

The authors do report an interesting approach to the electrochemical selective hydrogenation of acetylene but a lot of evidence on the active site structure and proposed mechanism is not very convincing. Overall, I think, however, that this manuscript could become publishable after careful revision.

Comments from Reviewer #2

The manuscript authored by Bai et al. presents a significant advancement in the domain of C₂H₄ production derived via electrocatalytic semi-hydrogenation of C₂H₂. The researchers drew inspiration from the CO₂RR field and devised a Cu-F/GDE electrode. This electrode achieved a substantial C₂H₂ to C₂H₄ conversion and high Faradaic efficiency. The efficacy of Cu-F was attributed to the enhanced water dissociation rate facilitated by F[•]. It is suggested that the H^{} was produced from Cu sites near F and spillover onto Cu sites far from F, where the hydrogenation of C₂H₂ takes place. The*

work showcased in the manuscript is intriguing as the authors conducted a comprehensive investigation from various standpoints encompassing catalyst design, device engineering, process optimization, and mechanistic elucidation through DFT and in situ Raman spectroscopy. Given its breadth and depth, this work appears suitable for publication in Nature Communications. Nevertheless, I would like to offer some comments and pose questions that could potentially enhance the manuscript's quality.

1. *There is a typo on Page 4, Line 103: It should be NH₄HF₂ instead of NH₄HF.*

Answer: Thank to the comments. In the revised manuscript, the typo of NH₄HF had been corrected to NH₄HF₂ in page 4 line 100. Moreover, we had carefully checked the manuscript to eliminate other typos.

“Typically, the preparation of the Cu-F involves treating Cu(NO₃)₂ and NH₄HF₂ in DMF via a solvent-thermal process”.

2. *For figures lacking a legend, it would be beneficial to provide additional contextual information within the caption. For instance, in Fig 2a, clarification regarding the conditions of the dashed and solid lines would be helpful.*

Answer: Thanks to the nice comment. A clarification regarding the conditions of the dashed and solid lines had been added for Figure 2a in the revised manuscript. We also provided additional contextual information within the caption for other figures lacking a legend. Such as:

“**Figure 2.** Electrocatalytic performance of Cu-F in 1 cm² flow cell. (a) LSV curves in Ar (dash) and 70% C₂H₂/Ar (sphere). (b) partial current density for C₂H₄ and H₂. (c) Faraday efficiency for C₂H₄ and H₂ at various potential and the corresponding current density. (d) C₂H₄ formation rate. (e) long-term stability at a constant current density of 200 mA cm⁻². All tests are measured using a three-electrode flow cell (1 cm²) in 1 M KOH at room temperature under 70 mol% C₂H₂/Ar flow (30 ml min⁻¹). The results are presented without iR compensation except e. For e the R=9.4 Ω.”

3. *The size of Figure 2b is insufficient. To improve visibility, relocating the inset image*

to the Supplementary Information and labeling the shift in onset potentials directly onto the plot could be considered.

Answer: Thanks to the nice comments. As the reviewer's suggestion, the inset image of Figure 2b had been relocated into the Supplementary Information as Figure S21. And the shift in onset potentials had been directly labeled onto the plot.

Figure 2b. Partial current density for C₂H₄ and H₂.

Figure S21. The onset potentials over Cu-F for C₂H₄ and H₂ generation. Measured using a three-electrode flow cell (1 cm²) in 1 M KOH at room temperature under 70 mol% C₂H₂/Ar flow (30 ml min⁻¹). The results are presented without iR compensation.

4. Could you please elaborate on the rationale behind reporting the system without iR compensation? Did the solution resistance exhibit significant variability across runs or with differing feed concentrations?

Answer: Thank you for the professional comments. The solution resistance had been

provided in the experimental section in the supporting information, which shows no significant differences across runs or with differing feed concentrations. Moreover, the solution resistance over Cu-F was relative smaller than that of Cu NP, thus we thought that the electrochemical results without iR compensation is clear enough to confirm the superior advantages of Cu-F in the electrochemical semi-hydration of C₂H₂.

“The R_u resistances at working conditions in electrode system are listed in the table below:

Electrode system	Catalysts	Resistances (Ω)	Electrode area (cm ²)
Three electrode system	Cu-F	1.1	1
	Cu-Cl	1.2	
	Cu-Br	1.2	
	Cu-I	1.2	
	Cu NP	1.2	
Two electrode system	Cu-F	9.4	
Three electrode system	Cu-F	1.5	25
Two electrode system	Cu-F	9.7	
Three electrode system	Cu-F	1.1 @50 mol % C ₂ H ₂	1
		1.1 @ 5 mol % C ₂ H ₂	

All the electrochemical performances are presented without the IR compensation, except the stability test in the 1 cm² flow cell (Figure 2e). As flow cell (1 cm²) stability test, the curvy was compensate at 200 mA with the solution resistance was about 9.4 Ω.”

5. Has the I-V curve been observed to alter when the gas flow rate is decreased from 30 to 6 mL/min?

Answer: Thank a lot for the nice comments. The LSV curves over Cu-F under the 70 mol% C₂H₂ flow at 6, 15 and 30 ml min⁻¹ had been recorded, respectively. The results were presented in the revised manuscript as Figure S18. As shown, the LSV curves showed no obvious changes, although the flow rate of the feed gas decreased from 30 to 6 ml min⁻¹. The effect of the flow rate on the LSV results had been also clarified in the revised manuscript at page 7 line 162-165:

“The single-pass C₂H₂ conversion over Cu-F reaches 30.3 %, which increases to 78.5%

when reducing the flow rate to 6 ml min⁻¹, and the LSV curves show no obvious change with the flow rate (Figure S18).”

Figure S18. ESAE performance of Cu-F at different flow rate of the feed gas. (a) Single-path C₂H₂ conversion vs. current density. (b) LSV curves. Measured using a three-electrode flow cell (1 cm²) in 1 M KOH at room temperature under 70 mol% C₂H₂/Ar flow. The results are presented without iR compensation.

6. The section pertaining to mechanistic insights is captivating. However, the descriptions of the key experiments appear rather concise and abrupt. It might be advantageous to expound upon this section. To facilitate comprehension, the following elaborations could be valuable:

Answer: Thank you very much, these comments would help us significantly improving the mechanistic investigation on the ESAE process. The comments were firstly responded point to point as below:

- Could you elucidate how the insensitivity of Faradaic efficiency to pH alterations excludes the plausibility of a PCET process, especially when the proton source is H₂O instead of H⁺?

Answer: Thanks. A typical electron-coupled proton transfer from the water layer for C₂H₂ hydronation follow the process of *C_xH_y + H₂O + e⁻ → *C_xH_{y+1} + OH⁻. As reported, this process is more favorable under relative lower pH, which may be explained by the chemical equilibrium. Therefore, a higher overall FE for C₂H₂ hydronation would be gained under lower pH than higher pH (Nature Commun., 2021, 12, 7072). However, in this present work, the pH showed insignificant effect on the FE of Cu-F, as shown in Figure S33. Thus, it can be inferred that the C₂H₂ is more likely be hydronated by the surface adsorbed H*, rather than via electron-coupled proton transfer from the water layer.

Figure S33. The effect of the KOH concentration on the FE over Cu-F. Measured using a three-electrode flow cell (1 cm²) at room temperature under 70 mol% C₂H₂/Ar flow (30 ml min⁻¹). The results are presented without iR compensation.

- Can you detail how *tert*-butanol functions as an H atom capturer? Were different reaction products observed upon the addition of *tert*-butanol?

Answer: Thanks. Tertiary butanol (*t*-BuOH) can scavenge H^{*} and convert it into inert 2-methyl-2-propanol radicals, thus it was widely used as a capturer to study the effect in of H^{*} (Environ. Sci. Technol. 2019, 53, 11932-11940; Angew. Chem. 2020, 132, 21356-21361; Chem. Eng. J. 2019, 359, 894-901.). In this work, the Cu-F displays inferior performance after the addition of *t*-BuOH, confirming the participation of surface H^{*} in the C₂H₂ hydrogenation process (Figure S35a). Moreover, the addition of *t*-BuOH showed no impact on the composition of products, no different reaction products were observed (Figure S35b).

Figure S35. (a) Potential-dependent C₂H₂ conversion change over Cu-F with or without the addition of *tert*-Butanol in the electrolyte. (b) FE of Cu-F with *tert*-Butanol in the electrolyte. Measured using a three-electrode flow cell (1 cm²) in 1 M KOH at room temperature under 70 mol% C₂H₂/Ar flow (30 ml min⁻¹). The results are presented without iR compensation.

- Regarding the KIE experiment, could you offer further clarification on how high KIE

values indicate that the H_2O dissociation is the rate-determining step? Hence, lower KIE values could signify improved water dissociation.

Answer: As shown in Figure 3c, the Cu NP displayed a high KIE value of 2.01, which is characteristic of primary KIE (Angew. Chem. Int. Edit. 2022, 61, e202206233). The primary KIE means the cleavage of the H-O/D-O bond limits the reaction, thus the water dissociation is involved in the RDS. On the other hand, the KIE value of Cu-F decreased significantly to 1.26, the water dissociation is no longer involved in the RDS, demonstrating that Cu-F promoted the water dissociation process.

Figure 3c. Kinetic isotope effect.

- The authors propose that H_2O dissociation transpires at the Cu-F near site, with H^* subsequently transferring to the Cu-F far site. Could you provide insight into the distance between individual F atoms? Additionally, have energy barriers for the spillover process been investigated?

Answer: Thank you very much. The distance between individual F atoms in Cu-F had been estimated based on the surface area, and surface F content. The results showed the surface density of F atom on Cu-F was about 3.2 per nm², thus the distance between two individual F atoms was estimated as 0.8 nm.

Additionally, the energy for the H transfer on the surface of Cu-F had been calculated using CINEB, according to the reviewer's comments. As shown in Figure S41, the transformation from IS to TS1 displayed the highest energy barrier of 0.105 eV, suggesting the transferring of H on the Cu-F surface is relatively easy.

Figure S41. Energy of the system as function of the transfer pathway for a surface *H on Cu-F, calculating based on CINEB.

- The distinct in situ Raman peaks corresponding to C=C on Cu-F and Cu NP raise questions. How did the authors verify that the peak at 1554 cm^{-1} indeed represents C=C? Furthermore, the doublet peaks between 1200-1600 appear akin to typical Id and Ig peaks of carbon substrates. Could you shed light on their differentiation?

Answer: Thanks. As reported, the Raman peaks for the D-peak and G-peak of carbon substrate were commonly detected at about 1350 and 1610 cm^{-1} (Nature Commoun., 2023, 14, 2137; Phys. Rev. B, 2001, 64, 7). In this work, these two peaks were not observed, which may due to the cover of Cu on the substrate. Moreover, the two dominant peaks at 1127 and 1513 cm^{-1} can be assigned to the C-C and C \equiv C vibrations of polyacetylene (Nature Commoun., 2023, 14, 2137; Nature Catal., 2021, 4, 557; J. Phys. Chem., 1985, 89, 5046), these two peaks had been clarified in the revised manuscript. The small peak at 1554 cm^{-1} , which besides the peak of 1513 cm^{-1} , is the characteristic signal of C=C stretch modes for π -bonded ethylene (Nature Catal., 2021, 4, 557; Nature Commoun., 2023, 14, 2137; Nature Sustain., 2023, 6, 827), therefore this peak of 1554 cm^{-1} represents the adsorbed ethylene on the Cu-F.

Figure 3a, b. Potential-dependent operando Raman spectra, measured using a three-electrode observable cell in 1 M KOH at room temperature under 70 mol% C_2H_2/Ar flow (30 ml min^{-1}).

Ultimately, based on the reviewer's comments, the mechanistic insights section had

been expounded in the revised manuscript to make the description more clearly.

Page 8 line 198-203: “As shown in Figure 3a, the two peaks at 1127 and 1513 cm^{-1} can be assigned to the C-C and C \equiv C vibrations of polyacetylene²². The signal belonging to the C \equiv C vibration is detected at $\sim 1700 \text{ cm}^{-1}$ at an open circuit potential (OCP), indicating the adsorption of acetylene on Cu-F¹⁸. This signal disappears when the potential increase to -0.4 V (vs. RHE), and new signals belonging to bonded ethylene emerge at ~ 1343 and $\sim 1554 \text{ cm}^{-1}$ ^{14,18}, which confirm the occurrence of EASE process on Cu-F.”

Page 8-9 line 212-228: “The pH of the electrolyte is investigated to reveal the hydrogen source in the EASE. As reported by Deng et al., a typical electron-coupled proton transfer process ($^*C_xH_y + H_2O + e^- \rightarrow ^*C_xH_{y+1} + OH^-$) is more favorable under relative low pH²⁰. However, the pH shows insignificant effect on the overall FE of Cu-F (Figure S32-34), which excludes the electron-coupled proton transfer process. The hydrogenation of C_2H_2 is preferentially facilitated by the surface absorbed *H , which stem from the water dissociation. The surface *H is further judged by adding of tert-Butanol (*t*-BuOH) into the electrolyte. The *t*-BuOH can capture *H to form inert 2-methyl-2-propanol radicals, leading to a suppressed hydrogenation activity³¹⁻³². As shown in Figure S35, the Cu-F displays inferior performance after adding *t*-BuOH, confirming the participation of surface *H in the hydrogenation of C_2H_2 . To gain insights into the role of water dissociation in *H generation, the kinetic isotope effect (KIE) of H/D is studied during EASE. The Cu NP shows a KIE of 2.01 (Figure 3c and Figure S36), which is characteristic of primary KIE, indicating the involvement of water dissociation in the rate-determining step (RDS)²⁹. For Cu-F, the KIE decrease to 1.26, water dissociation is no longer involved in the RDS, demonstrating that Cu-F accelerates the hydrogen transfer process via promoting water dissociation³³.”

7. Kindly elucidate the rationale behind designing the tandem 1-25 cm^2 device in such a configuration. What considerations guided this particular process design?

Answer: Thanks to the comments. The designing of the tandem 1-25 cm^2 device was based on the experimental results, with the purpose of directly converting 70 mol% C_2H_2 to C_2H_4 . We firstly revealed the efficient ESAE performance of Cu-F in the 1 cm^2 flow cell, proved by the superior current density, C_2H_4 FE, C_2H_4 formation rate and stability (Figure 1). Moreover, as shown in Figure S18, the single-path C_2H_2 conversion reached about 80% under a flow rate of 6 ml min^{-1} in the 1 cm^2 flow cell, reducing the C_2H_2 concentration from 70 mol % to about 14 mol %. This demonstrated the capacity of the 1 cm^2 flow cell to deal with C_2H_2 with high concentration. However, the 1 cm^2 flow cell show insufficient ability with low C_2H_2 concentration. As shown in Figure S44, when the C_2H_2 concentration was reduced to 1 mol%, the C_2H_4 FE in the 1 cm^2 flow cell is merely about 20%. On other hand, the 25 cm^2 flow cell with s-type chamber

was proved competent to deal with the 1 mol% C₂H₂ by the notable conversion and selectivity (Figure 4a-b and Figure S46), even in the C₂H₄ environment. Therefore, the tandem 1-25 cm² device used in this work was designed for combining the capacity of the 1 cm² flow cell to deal with high concentration C₂H₂ feed gas, and the ability of the 25 cm² flow cell for converting residual C₂H₂ with a low concentration. The designing of the tandem 1-25 cm² device had been clarified in the revised manuscript at page 12, line 301-303:

“The designing is based on the capacity of the 1 cm² flow cell to deal with high concentration C₂H₂ feed gas, and the ability of the 25 cm² flow cell for converting residual C₂H₂ at low concentration.”

Figure S18. ESAE performance of Cu-F at different flow rate of the feed gas. (a) Single-path C₂H₂ conversion vs. current density. (b) LSV curves. Measured using a three-electrode flow cell (1 cm²) in 1 M KOH at room temperature under 70 mol% C₂H₂/Ar flow. The results are presented without iR compensation.

Figure S44. ESAE performance of Cu-F under C₂H₂/Ar flow with different C₂H₂ concentration (mol%, 30 ml min⁻¹). (a) LSV curves. (b) Faraday efficiency and current density at different C₂H₂ concentration. Measured using a three-electrode flow cell (1 cm²) in 1 M KOH at room temperature. The results are presented without iR compensation.

Figure 4. (a) C₂H₂ conversion and (b) long-term operation of EASE over Cu-F under ethylene-rich environment, measured using a three-electrode flow cell (25 cm²) in 1 M KOH at room temperature under 1 mol% C₂H₂/C₂H₄ flow (20 ml min⁻¹), constant current density set as 40 mA cm⁻².

Figure S46. Selectivity vs. time of Cu-F in a 25 cm² flow-cell. Measured using a three-electrode flow cell (25 cm²) in 1 M KOH at room temperature under 1 mol% C₂H₂/C₂H₄ flow (20 ml min⁻¹). A constant current density is set as 40 mA cm⁻². The results are presented without iR compensation.

8. *The demonstration of stability over a mere 3000s appears rather brief. What intrinsic limitations influence this choice of duration?*

Answer: Thanks for your comments. We lengthen the long-term operation time on the tandem device to 3h, and the EASE performance of Cu-F showed no significant decay. The result had been presented at Figure 4d in the revised manuscript.

Figure 4d. (d) Long-term operation of ESAE in the tandem flow cells, 70 mol% C₂H₂ in Ar, flow rate 6 ml min⁻¹, constant potentials set as -1.1 and -0.6 V (vs. RHE) for 1 cm² and a 25 cm² flow cells, respectively.

REVIEWERS' COMMENTS

Reviewer #1 (Remarks to the Author):

The authors have undertaken considerable efforts to strengthen the claims presented in the manuscript and I can now recommend this manuscript for publication in Nat. Commun.

Reviewer #2 (Remarks to the Author):

The authors have done an excellent job addressing all my questions and suggestions. Just couple of minor points found in this version of manuscript though:

- 1) Fig 2e, 2-electrode cell potential should not be iR compensated. To prevent deception, perhaps they might rename the y-label to something else.
- 2) Fig S5, F 1s of Cu-F isn't well-fitted (the fitted curve is too small compared to the data).
- 3) Mixed use of "EASE" and "ESAE" throughout the entire manuscript.

Reviewer #1 (Remarks to the Author):

The authors have undertaken considerable efforts to strengthen the claims presented in the manuscript and I can now recommend this manuscript for publication in Nat. Commun.

Answer: Thank you for the nice comments.

Reviewer #2 (Remarks to the Author):

The authors have done an excellent job addressing all my questions and suggestions. Just couple of minor points found in this version of manuscript though:

1. Fig 2e, 2-electrode cell potential should not be *iR* compensated. To prevent deception, perhaps they might rename the y-label to something else.

Answer: Thanks to the comments. According to the reviewer's comment, the y-label of Figure 2e had been renamed as E_{cell} with *iR* compensation.

Figure 2. (e) long-term stability at a constant current density of 200 mA cm^{-2} . All tests are measured using a three-electrode flow cell (1 cm^2) in 1 M KOH at room temperature under $70 \text{ mol\% C}_2\text{H}_2/\text{Ar}$ flow (30 ml min^{-1}). The results are presented without *iR* compensation except e. For e the $R=9.4 \Omega$.

2. Fig S5, F 1s of Cu-F isn't well-fitted (the fitted curve is too small compared to the data).

Answer: Thanks to the comment. The fitting of F 1s XPS spectrum for Cu-F had been revised, the new results fitted well with the experimental results.

Figure S5. F 1s XPS spectra of Cu-F and Cu(OH)F.

3. *Mixed use of “EASE” and “ESAE” throughout the entire manuscript.*

Answer: Thanks. In the revised manuscript, the related acronym had been unified as ESAE.